# Latent Equilibrium: A unified learning theory for arbitrarily fast computation with arbitrarily slow neurons

**Paul Haider**[1]  **Benjamin Ellenberger**[1]  **Laura Kriener**[1]
**Jakob Jordan**[1]  **Walter Senn**[1,*]  **Mihai A. Petrovici**[1,2,*]

[1]Department of Physiology, University of Bern
[2]Kirchhoff-Institute for Physics, Heidelberg University

{first_name}.{surname}@unibe.ch

## Abstract

The response time of physical computational elements is finite, and neurons are no exception. In hierarchical models of cortical networks each layer thus introduces a response lag. This inherent property of physical dynamical systems results in delayed processing of stimuli and causes a timing mismatch between network output and instructive signals, thus afflicting not only inference, but also learning. We introduce Latent Equilibrium, a new framework for inference and learning in networks of slow components which avoids these issues by harnessing the ability of biological neurons to phase-advance their output with respect to their membrane potential. This principle enables quasi-instantaneous inference independent of network depth and avoids the need for phased plasticity or computationally expensive network relaxation phases. We jointly derive disentangled neuron and synapse dynamics from a prospective energy function that depends on a network's generalized position and momentum. The resulting model can be interpreted as a biologically plausible approximation of error backpropagation in deep cortical networks with continuous-time, leaky neuronal dynamics and continuously active, local plasticity. We demonstrate successful learning of standard benchmark datasets, achieving competitive performance using both fully-connected and convolutional architectures, and show how our principle can be applied to detailed models of cortical microcircuitry. Furthermore, we study the robustness of our model to spatio-temporal substrate imperfections to demonstrate its feasibility for physical realization, be it in vivo or in silico.[§]

## 1 Introduction

Physical systems composed of large collections of simple, but intricately connected elements can exhibit powerful collective computational properties. A prime example are animals' nervous systems, and most prominently the human brain. Its computational prowess has motivated a large, cross-disciplinary and ongoing endeavor to emulate aspects of its structure and dynamics in artificial substrates, with the aim of being ultimately able to replicate its function. The speed of information processing in such a system depends on the response time of its components; for neurons, for example, it can be the integration time scale determined by their membrane time constant.

---

[*]Senior authors.
[§]Code is available at https://github.com/unibe-cns/le_NeurIPS_code.

35th Conference on Neural Information Processing Systems (NeurIPS 2021).

If we consider hierarchically organized neural networks composed of such elements, each layer in the hierarchy causes a response lag with respect to a changing stimulus. This lag introduces two related critical issues. For one, the inference speed in these systems decreases with their depth. In turn, this induces timing mismatches between instructive signals and neural activity, which disrupts learning. For example, recent proposals for bio-plausible implementations of error backpropagation (BP) [1–4] in the brain all require some form of relaxation, both for inference and during learning [5–11]. Notably, this also affects some purely algorithmic methods involving auxiliary variables [12]. To deal with this inherent property of physical dynamical systems, two approaches have been suggested: either phased plasticity that is active only following a certain relaxation period, or long stimulus presentation times with small learning rates. Both of these solutions entail significant drawbacks: the former is challenging to implement in asynchronous, distributed systems such as cortical networks or neuromorphic hardware, while the latter results, by construction, in slow learning. This has prompted the critique that any algorithm requiring such a settling process is too slow to describe complex brain function, particularly when involving real-time responses [13]. To the best of our knowledge this fundamental problem affects all modern models of approximate BP in biological substrates [5–11].

To overcome these issues, we propose a novel framework for fast computation and learning in physical substrates with slow components. As we show below, this framework jointly addresses multiple aspects of neuronal computation, including neuron morphology, membrane dynamics, synaptic plasticity and network structure. In particular, it provides a biologically plausible approximation of BP in deep cortical networks with continuous-time, leaky neuronal dynamics and local, continuous plasticity. Moreover, our model is easy to implement in both software and hardware and is well-suited for distributed, asynchronous systems.

In our framework, inference can be arbitrarily fast (up to finite simulation resolution or finite communication speed across physical distances) despite a finite response time of individual system components; downstream responses to input changes thus become effectively instantaneous. Conversely, responses to instructive top-down input that generate local error signals are also near-instantaneous, thus effectively removing the need for any relaxation phase. This allows truly phase-free learning from signals that change on much faster time scales than the response speed of individual network components.

Similarly to other approaches [5, 6, 9, 14, 15], we derive neuron and synapse dynamics from a joint energy function. However, our energy function is designed to effectively disentangle these dynamics, thus removing the disruptive co-dependencies that otherwise arise during relaxation. This is achieved by introducing a simple, but crucial new ingredient: neuronal outputs that try to guess their future state based on their current information, a property we describe as "prospective" (which should not be confused with the "predictive" in predictive coding, as we also discuss below). Thereby, our framework also constructs an intimate relationship between such "slow" neuronal networks and artificial neural network (ANN)[1], thus enabling the application of various auxiliary methods from deep learning.

## 2  The problems of slow components

To illustrate the issue with relaxation, we consider two neurons arranged in a chain (Fig. 1a). Biological neuronal membrane potentials $u$ are conventionally modeled as leaky integrators of their input $I$: $C_\mathrm{m}\dot{u} = -g_\mathrm{l}u + I$, where the membrane capacitance $C_\mathrm{m}$ and leak conductance $g_\mathrm{l}$ determine the membrane time constant $\tau^\mathrm{m} := C_\mathrm{m}/g_\mathrm{l}$ and thereby its response speed. These dynamics attenuate and delay input, resulting in significant differences between neuronal output rates and those expected in an instantaneous system (Fig. 1b). This mismatch increases with every additional layer: a feedforward network with $n$ layers has an effective relaxation time constant of approximately $n\tau^\mathrm{m}$.

Besides slow inference, this delayed response leads to critical issues during learning from downstream instructive signals. Consider the typical scenario where such a target signal is present in the output layer and plasticity is continuously active (and not phased according to some complicated schedule). If the system fulfills its task correctly, the target signal corresponds to the output of the relaxed system and no learning should take place. However, due to delays in neuronal responses, the output signal differs from the target during relaxation, which causes plasticity to adapt synaptic weights in an effort to better match the target. As a result, the system "overshoots" during early relaxation and has to

---

[1]To differentiate between biologically plausible, leaky neurons and abstract neurons with instantaneous response, we respectively use the terms "neuronal" and "neural".

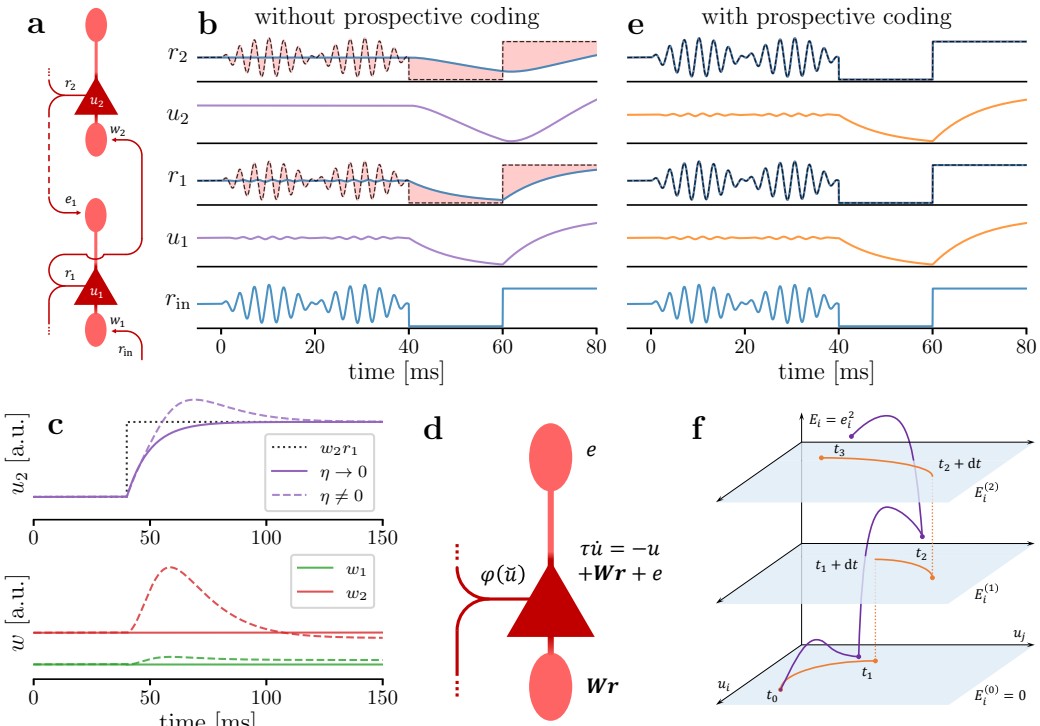

**Figure 1: Prospective coding solves the relaxation problem.** **(a)** A simple, functionally feedforward network of two neurons. Note the recurrence induced by the propagation of both bottom-up signals and top-down errors. **(b)** Neuronal firing rates (blue) and membrane potentials (purple) for an input sequence consisting of a quickly but smoothly changing stimulus followed by two constant stimuli. Dashed black lines denote instantaneous (relaxed) rates. Red shading highlights the mismatch between instantaneous and actual firing rates that disrupts both inference and learning. **(c)** Continuous learning during relaxation without prospective coding in the network from (a). Dotted black: target membrane potential. Solid lines: trajectories for vanishing learning rates. Dashed lines: trajectories for nonzero learning rates. Purple: membrane potentials. Note the overshoot when learning is continuously active. Green/red: presynaptic weights of 1st/2nd neuron. **(d)** Sketch of neuron model derived from Latent Equilibrium (LE). The membrane potential still reacts slowly to input, but the output firing rate uses the prospective membrane potential $\breve{u}^{\mathrm{m}}$. **(e)** Same as (b), but with LE. Note the instantaneous reaction to input changes of the second neuron. **(f)** Sketch of the mismatch energy $E_i$ for a hidden neuron during learning for an input with several jumps (for clarity, we assume the jumps to be upwards for consecutive inputs). In LE (orange), the energy itself jumps with the input, but then remains constant while neuron dynamics $(u_i, u_j)$ evolve. Without LE (purple), the energy changes transiently and plasticity follows incorrect gradients before relaxation. The trajectories for $E = 0$ correspond to the scenario from (c). Planes of constant energy are drawn for visual guidance and do not represent constant-energy manifolds.

undo these synaptic changes through further learning in the late relaxation phase (Fig. 1c). Also, since output errors need inputs to propagate forward through the entire network before propagating backwards themselves, the input layer only observes correct errors after about $2n\tau^{\mathrm{m}}$. We refer to this issue as the "relaxation problem". In the following, we present a solution for this problem, combining prospective neuron dynamics with continuously active, local synaptic plasticity.

## 3  Fast computations in slow substrates

Inspired by the canonical coordinates from classical mechanics, we describe the state of a neuronal network by its position in phase space $(\boldsymbol{u}, \dot{\boldsymbol{u}})$, with the generalized position $\boldsymbol{u}$ and the generalized momentum $\dot{\boldsymbol{u}}$. The relation between these two components describes the physics of the system. To obtain the leaky integrator dynamics that characterize biological neurons, we first define the abstract network state $\breve{\boldsymbol{u}}^{\mathrm{m}}$ as

$$\breve{\boldsymbol{u}}^{\mathrm{m}} := \boldsymbol{u} + \tau^{\mathrm{m}}\dot{\boldsymbol{u}} \ . \tag{1}$$

We next define an energy function $E$ that characterizes this state:

$$E(\breve{\boldsymbol{u}}^{\mathrm{m}}) := \frac{1}{2} \left\| \breve{\boldsymbol{u}}^{\mathrm{m}} - \boldsymbol{W}\boldsymbol{\varphi}(\breve{\boldsymbol{u}}^{\mathrm{m}}) - \boldsymbol{b} \right\|^2 + \beta\mathcal{L}(\breve{\boldsymbol{u}}^{\mathrm{m}}) \ . \tag{2}$$

Here, $\boldsymbol{W}$ and $\boldsymbol{b}$ represent the weight matrix and bias vector, $\boldsymbol{\varphi}$ is the neuronal activation function and the loss $\mathcal{L}$ is scaled by a constant $\beta$; we use bold font for matrices, vectors, and vector-valued functions. Intuitively speaking, $E$ measures the difference between "what neurons guess that they will be doing in the future" ($\breve{\boldsymbol{u}}^{\mathrm{m}}$) and "what their biases and presynaptic afferents believe they should be doing" ($\boldsymbol{b} + \boldsymbol{W}\boldsymbol{\varphi}(\breve{\boldsymbol{u}}^{\mathrm{m}})$). Furthermore, for a subset of neurons, it adds a loss $\mathcal{L}$ that provides an external measure of the error for the states of these neurons, determined by instructive signals such as labels, reconstruction errors or rewards. This formulation of the energy function is very generic and can, by appropriate choice of parameters, apply to different network topologies, including multilayer perceptrons, convolutional architectures and recurrent networks. Note that this energy can be written as a neuron-wise sum over mismatch energies plus, for factorizing loss functions, a local loss term $E_i(\breve{u}_i^{\mathrm{m}}, \boldsymbol{r}_{i,\mathrm{pre}}) = \frac{1}{2}(\breve{u}_i^{\mathrm{m}} - \boldsymbol{W}_i \boldsymbol{r}_{i,\mathrm{pre}} - b_i)^2 + \beta\mathcal{L}_i(\breve{u}_i^{\mathrm{m}})$, where $\boldsymbol{r}_{i,\mathrm{pre}} := \boldsymbol{\varphi}(\breve{\boldsymbol{u}}_{i,\mathrm{pre}}^{\mathrm{m}})$ is the presynaptic input vector for the $i$th neuron in the network.

We can now derive neuronal dynamics as extrema of the energy function from Eqn. 2:

$$\nabla_{\breve{\boldsymbol{u}}^{\mathrm{m}}} E = 0 \quad \Longrightarrow \quad \tau^{\mathrm{m}}\dot{\boldsymbol{u}} = -\boldsymbol{u} + \boldsymbol{W}\boldsymbol{\varphi}(\breve{\boldsymbol{u}}^{\mathrm{m}}) + \boldsymbol{b} + \boldsymbol{e} \ , \tag{3}$$

with presynaptic bottom-up input $\boldsymbol{W}\boldsymbol{\varphi}(\breve{\boldsymbol{u}}^{\mathrm{m}})$ and top-down error signals

$$\boldsymbol{e} = \boldsymbol{\varphi}'(\breve{\boldsymbol{u}}^{\mathrm{m}})\boldsymbol{W}^{\mathrm{T}}\left[\breve{\boldsymbol{u}}^{\mathrm{m}} - \boldsymbol{W}\boldsymbol{\varphi}(\breve{\boldsymbol{u}}^{\mathrm{m}}) - \boldsymbol{b}\right] \tag{4}$$

for hidden neurons and $\boldsymbol{e} = -\beta\nabla_{\breve{\boldsymbol{u}}^{\mathrm{m}}}\mathcal{L}$ for neurons which directly contribute to the loss. Plugging Eqn. 3 into Eqn. 4, it is easy to see that in hierarchical networks these errors can be expressed recursively over layers $\ell$, $\boldsymbol{e}_\ell = \boldsymbol{\varphi}'(\breve{\boldsymbol{u}}^{\mathrm{m}})\boldsymbol{W}_{\ell+1}^{\mathrm{T}}\boldsymbol{e}_{\ell+1}$, thus instantiating a variant of BP. Eqn. 3 can be interpreted as the dynamics of a structured pyramidal neuron receiving presynaptic, bottom-up input via its basal dendrites and top-down input via its apical tree (Fig. 1d). We provide a more detailed description of our model's biophysical implementation in Section 5. We note that our approach is in contrast to previous work that introduced neuron dynamics via gradient descent on an energy function, such as [5, 6, 16], whereas we require a stationary energy function with respect to $\breve{u}^{\mathrm{m}}$. Indeed, this difference is crucial for solving the relaxation problem, as discussed below. Since for a given input our network moves, by construction, within a constant-energy manifold, we refer to our model as Latent Equilibrium (LE).

We can now revisit our choice of $\breve{u}^{\mathrm{m}}$ from a functional point of view. Instead of the classical output rate $\varphi(u)$, our neurons fire with $\varphi(\breve{u}^{\mathrm{m}})$, which depends on both $u$ and $\dot{u}$ (Eqn. 1). As neuron membranes are low-pass filters (Eqn. 3), $\breve{u}^{\mathrm{m}}$ can be viewed as a prospective version of $u$: when firing, the neuron uses its currently available information to forecast the state of its membrane potential after relaxation. The prospective nature of $\breve{u}^{\mathrm{m}}$ also holds in a strict mathematical sense: the breve operator $\breve{}^{\mathrm{m}} := (1 + \tau^{\mathrm{m}}\mathrm{d}/\mathrm{d}t)$ is the exact inverse of an exponential low-pass filter (see SI). While neuronal membranes continue to relax slowly towards their steady states, neuronal outputs use membrane momenta to compute a correct instantaneous reaction to their inputs, even in the case of jumps (Fig. 1e). Thus, information can propagate instantaneously throughout the network, similarly to an ANN, counterintuitively even when membrane dynamics are never in equilibrium. The activity of the output layer hence reflects arbitrarily fast changes in the input – even on time scales smaller than the neuronal time constant $\tau^{\mathrm{m}}$ – rather than responding with a significant time lag and attenuation as in previous, gradient-based models.

The idea of incorporating derivatives into the input-output function of a system has a long history in control theory [17] and also represents a known, though often neglected feature of (single) biological neurons [18, 19]. A related, but different form of neuronal prospectivity has also been considered in other models of bio-plausible BP derived from a stationary action [20, 21]. At the level of neuronal populations with additive Gaussian noise, there exists a long tradition of studying faster-than-$\tau^{\mathrm{m}}$ responses, both with [22] and without [23] recurrent connectivity. Similar observations also hold for single neurons in the presence of noise [24, 25]. Building on these insights and integrating them into a unified theory of neuronal dynamics and learning, our model proposes a specific form of prospective coding that can also be learned by local adaptation mechanisms, as we discuss in Section 6.

We should also stress the difference between the terms "prospective" and "predictive". Predictive coding, or more generally, predictive processing, is a theory of brain function which proposes that

brains maintain internal models of their environment which they update and learn by trying to predict sensory input at the same moment in time. Originally based on a specific Bayesian model and error communication scheme [14], the notion of predictive coding can be generalized to layers in a hierarchical model predicting the activities of subsequent layers. This principle is instantiated in our networks as well, as will become clear in the following sections. In contrast, prospective coding in our framework refers to the ability of neurons to "look forward in time" using the current state of the membrane potential in phase space (position and velocity), as described above.[2] Prospective and predictive coding are thus complementary: our specific prospective mechanism provides a cure for relaxation in predictive coding networks, thus significantly speeding up both learning and inference, as we describe below.

Note that, even for functionally feedforward networks, our resulting network structure is recurrent, with backward coupling induced by error inputs to the apical tree. As a non-linear recurrent network, it cannot settle instantaneously into the correct state; rather, in numerical simulations, it jumps quickly towards an estimated stationary activity state and reaches equilibrium within several such jumps (of infinitesimal duration). In practice, saturating activation functions can help avoid pathological behavior under strong coupling. Moreover, we can introduce a very short exponential low-pass filter $\tau^s$ on top-down signals, slightly larger than the temporal resolution of the simulation. Thus, in physical systems operating in continuous time, $\tau^s$ can effectively become infinitesimal as well and does not affect the speed of information propagation through the network. In particular, as we discuss below, the perpetual consistency between input and output allows our model to continuously learn to reduce the loss, obviating the need for network relaxation phases and the associated global control of precisely timed plasticity mechanisms.

## 4 Fast learning in slow substrates

Based on our prospective energy function (Eqn. 2), we define synaptic weight dynamics, i.e., learning, as time-continuous stochastic gradient descent with learning rate $\eta_W$:

$$\dot{\boldsymbol{W}} \propto -\nabla_{\boldsymbol{W}} E \quad \Longrightarrow \quad \dot{\boldsymbol{W}} = \eta_W \left[ \breve{\boldsymbol{u}}^{\mathrm{m}} - \boldsymbol{W}\boldsymbol{r} - \boldsymbol{b} \right] \boldsymbol{r}^{\mathrm{T}} . \tag{5}$$

Thus, weights evolve continuously in time driven by local error signals without requiring any particular schedule. Neuronal biases are adapted according to the same principle. Note that this rule only uses quantities that are available at the locus of the synapse (see also Section 5). Intuitively, this locality is enabled by the recurrent nature of the network: errors in the output units spread throughout the system, attributing credit locally through changes in neuronal membrane potentials. These changes are then used by synapses to update their weight in order to reduce the network loss. However, our learning rule is not an exact replica of BP, although it does approximate it in the limit of infinitely weak supervision $\beta \to 0$ (often referred to as "nudging"); strictly speaking, it minimizes the energy function $E$, which implicitly minimizes the loss $\mathcal{L}$. This form of credit assignment can be related to previous models which similarly avoid a separate artificial backward pass (as necessary in classical BP) by allowing errors to influence neuronal activity [27]. Plasticity in the weights projecting to output neurons depends on the choice of $\mathcal{L}$; for example, for an $L^2$ loss, plasticity in the output layer corresponds to the classical delta rule [28]: $\dot{\boldsymbol{W}}_N = \eta_W \beta \left[ \boldsymbol{r}_N^* - \boldsymbol{r}_N \right] \boldsymbol{r}_{N-1}^{\mathrm{T}}$.

Despite similarities to previous work, learning in our framework does not suffer from many of the shortcomings that we have already noted. Since activity propagates quasi-instantaneously throughout the network, our plasticity can be continuously active without disrupting learning performance. This is true by construction and most easily visualized for a sequence of (piecewise constant) input patterns: following a change in the input, membrane dynamics take place in a constant-energy manifold (Eqn. 3) across which synaptic weight dynamics remain unchanged, i.e., they equally and simultaneously pull downward on all points of this manifold (Fig. 1f). This disentangling of membrane and synaptic weight dynamics constitutes the crucial difference to previous work, where the neuronal mismatch energies $E_i$ change as dynamics evolve and thus can not represent the true errors in the network before reaching a fixed point. We further note that LE also alleviates the problem of unlearning in these other models: due to the misrepresentation of errors during relaxation, continuous learning

---

[2]A similar concept has also been discussed in [26], but with the aim of implementing the future discounted states required for temporal difference learning; this form of prospectivity thus addresses a different problem and results in very different neuronal and synaptic dynamics.

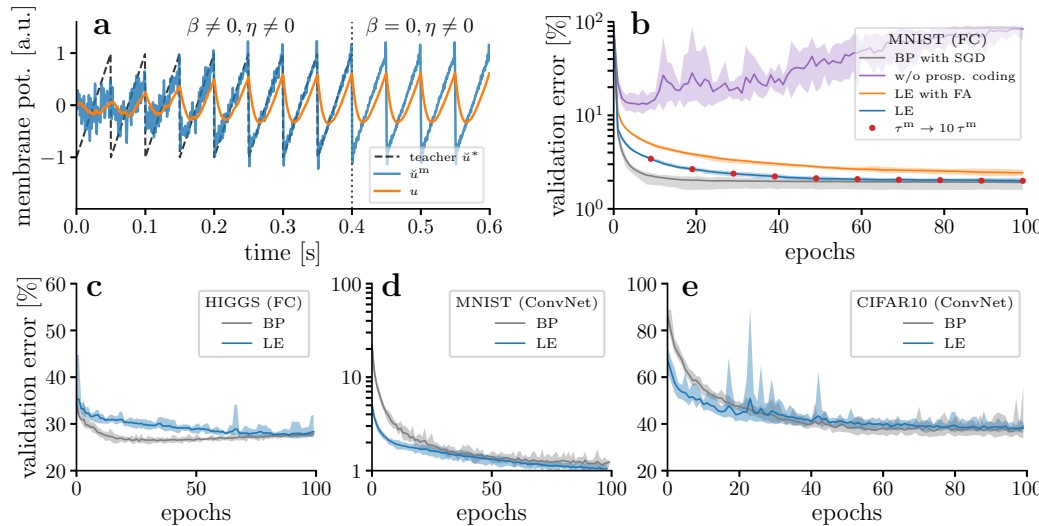

**Figure 2: Learning with LE. (a)** Fourier synthesis with fully connected (FC) network of size 50-30-1, receiving sinusoidal inputs with equidistant frequencies between 10 and 510 Hz. Teaching signal only present for the first 0.4 s, plasticity is continuously active. **(b)** MNIST dataset with FC network of size 784-300-100-10, presentation times $T_{\text{pres}} = 1\,\text{ms} = 0.05\,\tau^{\text{m}}$ per sample. Since variability for runs with different $\tau^{\text{m}}$ is statistically indistinguishable, we only show one set of corresponding min/max values. For comparison, we also show the performance with feedback alignment (FA) [31]. **(c)** HIGGS dataset with FC network of size 28-300-300-300-1. **(d)** MNIST with convolutional network (ConvNet) LeNet-5 [32]. **(e)** CIFAR-10 dataset (ConvNet: LeNet-5). For all examples, standard ANNs with the same topology trained with classical BP are shown for comparison.

changes synaptic weights even in perfectly trained networks (Fig. 1c). This disruption of existing memories is related, but not identical to catastrophic forgetting.

To illustrate time-continuous learning on continuous data streams, we first consider a simple Fourier-synthesis task in which a network driven by oscillating inputs learns to approximate a periodic (here: sawtooth) target signal. Despite the wavelengths of both input and target being much smaller than the time constant of the neuron, continuous plasticity in our model allows the output neuron to approximate the target well (Fig. 2a). The remaining differences merely reflect the fundamental limitation of a finite set of input frequencies and hidden units.

We next turn the more traditional paradigm of classification tasks to demonstrate the scalability of our model. We train feedforward networks using LE on several standard benchmark datasets (see SI for details) using sample presentation times much shorter than the membrane time constant. We first learn MNIST [29] and HIGGS [30] with a fully connected architecture. After 100 epochs, our model reaches classification test errors (mean $\pm$ std) of $(1.98 \pm 0.11)\,\%$ (MNIST) and $(27.6 \pm 0.4)\,\%$ (HIGGS), on par with a standard ANN trained using stochastic gradient descent (SGD) and reaching $(1.93 \pm 0.14)\,\%$ (MNIST) and $(27.8 \pm 0.4)\,\%$ (HIGGS) with the same architecture, i.e., with the same number of learnable parameters (Fig. 2b,c). With feedback alignment (FA, [31]), we achieve test errors of $(2.6 \pm 0.1)\,\%$ for MNIST. To illustrate the indifference of LE with respect to neuronal time constants, we repeated the MNIST experiments with significantly slower neuronal dynamics, obtaining the same results. In contrast, a network following the classical paradigm without prospective rates performs poorly: for membrane time constants of $\tau^{\text{m}} = 10\,\text{ms}$ and presentation times of $T = 100\,\text{ms}$ per sample, the network does not exceed 90% accuracy on MNIST. For even shorter presentation times, the performance of such models quickly degrades further.

Since our model does not assume any specific connectivity pattern, we can easily integrate different network topologies. Here we demonstrate this by introducing convolutional architectures on both MNIST and the more challenging CIFAR10 [33] datasets. On these datasets, our LE networks achieve test errors of $(1.1 \pm 0.1)\,\%$ (MNIST) and $(38.0 \pm 1.3)\,\%$ (CIFAR10), again on par with ANNs with identical structure at $(1.08 \pm 0.07)\,\%$ (MNIST) and $(39.4 \pm 5.6)\,\%$ (CIFAR10) (Fig. 2d,e). These results show that LE enables time-continuous learning using arbitrarily short presentation times in networks of leaky integrator neurons to achieve results competitive with standard BP in ANNs.

# 5 Fast computation and learning in cortical microcircuits

Due to the simplicity of their implementation, the principles of LE can be applied to models of approximate BP in the brain in order to alleviate the issues discussed above. Here we demonstrate how a network of hierarchically organized dendritic microcircuits [8, 34] can make use of our theoretical advances to significantly increase both inference and training speed, thus removing several critical shortcomings towards its viability as a scalable model of cortical processing. The resulting dynamical system represents a detailed and biologically plausible version of BP, with real-time dynamics, and phase-free, continual local learning able to operate on effectively arbitrary sensory timescales.

The fundamental building block of this architecture is a cortical microcircuit model consisting of pyramidal cells and interneurons (Fig. 3a,b). Pyramidal cells have three compartments: a basal dendrite receiving bottom-up input from lower areas, an apical dendrite receiving top-down input from higher areas and lateral input from interneurons, and a somatic compartment that integrates dendritic information and generates the neuronal output. Interneurons consist of two compartments: a basal dendrite receiving input from pyramidal cells in the same layer, and a somatic compartment that receives input from pyramidal cells in higher layers. Pyramidal somatic compartments are leaky integrators of input from neighboring compartments (see SI for the full set of equations):

$$C_{\mathrm{m}}\dot{u}_i^{\mathrm{som}} = g_{\mathrm{l}}\left(E_{\mathrm{l}} - u_i^{\mathrm{som}}\right) + g^{\mathrm{bas}}\left(v_i^{\mathrm{bas}} - u_i^{\mathrm{som}}\right) + g^{\mathrm{api}}\left(v_i^{\mathrm{api}} - u_i^{\mathrm{som}}\right) , \tag{6}$$

where $i$ is the neuron index, $E_{\mathrm{l}}$ the leak potential, $v_i^{\mathrm{bas}}$ and $v_i^{\mathrm{api}}$ the basal and apical membrane potentials, respectively, and $g^{\mathrm{bas}}$ and $g^{\mathrm{api}}$ the dendro-somatic couplings. Due to the conductance-based interaction between compartments, the effective time constant of the soma is $\tau^{\mathrm{eff}} := C_{\mathrm{m}}/(g_{\mathrm{l}} + g^{\mathrm{bas}} + g^{\mathrm{api}})$. For somatic membrane potentials, assuming that apical dendrites encode errors (see below) and basal dendrites represent the input, this corresponds directly to Eqn. 3. Following [35], plasticity in basal synapses is driven by the local error signal given by the discrepancy between

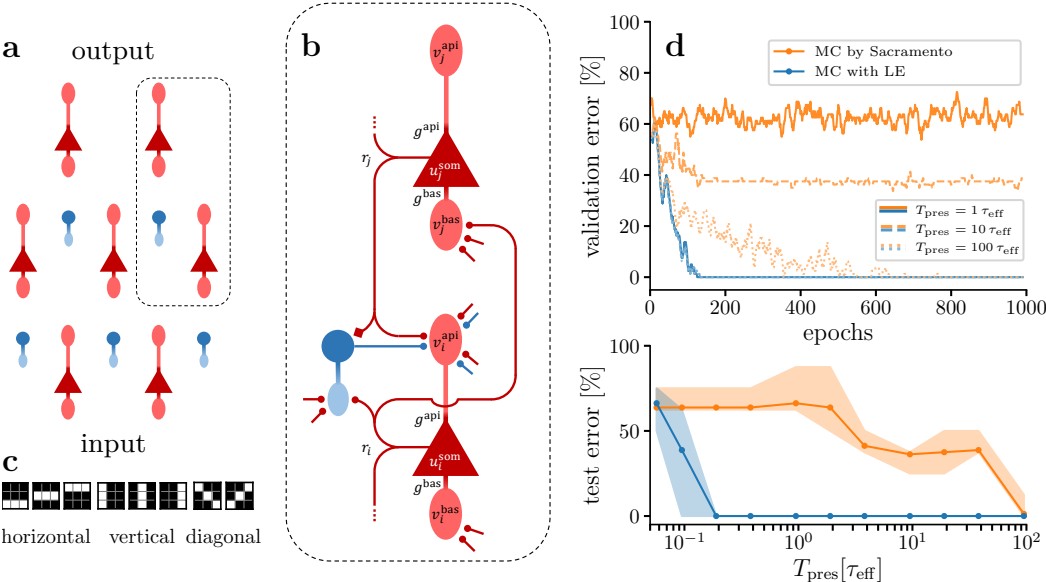

**Figure 3: LE in cortical microcircuits. (a,b)** Model architecture following [8]. Red: pyramidal cells, blue: interneurons; somatic compartments have darker colors. Each soma sends out a single axon that transmits either $\varphi(u)$ (Sacramento et al.) or $\varphi(\breve{u}^{\mathrm{m}})$ (LE). Except for top-down synapses, all synapses are plastic. **(c)** Synthetic dataset with 8 images grouped in 3 classes used to train a 3-layer network with 9-30-3 pyramidal cells. **(d)** Model performance during (top) and after (bottom) learning with (blue) and without (orange) LE. Top: Note the similarity in performance gains at the beginning of training, before the disruptive effects of relaxation begin to dominate. For better visualization, fluctuations are smoothed with a sliding window over 10 epochs. Bottom: Model performance (min, max and mean over 10 seeds) after 1000 epochs for different input presentation times.

somatic and dendric membrane potentials $u_i^{\text{som}}$ and $v_i^{\text{bas}}$:

$$\dot{w}_{ij} = \eta_W \left[ \varphi(u_i^{\text{som}}) - \varphi(\alpha v_i^{\text{bas}}) \right] \varphi(u_j^{\text{som}}) , \tag{7}$$

which is analogous to Eqn. 5 up to a monotonic transformation on the voltages (see SI).

In this architecture, plasticity serves two purposes. For pyramidal-to-pyramidal feedforward synapses, it implements error-correcting learning as a time-continuous approximation of BP. For pyramidal-to-interneuron synapses, it drives interneurons to mimic their pyramidal partners in the layers above (see also SI). Thus, in a well-trained network, apical compartments of pyramidal cells are at rest, reflecting zero error, as top-down and lateral inputs cancel out. When an output error propagates through the network, these two inputs can no longer cancel out and their difference represents the local error $e_i$. This architecture does not rely on the transpose of the forward weight matrix, improving viability for implementation in distributed asynchronous systems. Here, we keep feedback weights fixed, realizing a variant of feedback alignment. In principle, these weights could also be learned in order to further improve the local representation of errors Section 7.

Incorporating the principles of LE is straightforward and requires only that neurons fire prospectively: $\varphi(u) \rightarrow \varphi(\breve{u}^{\text{eff}})$. While we have already addressed the evidence for prospective neuronal activity, we note that our plasticity also uses these prospective signals, which constitutes an interesting prediction for future in-vivo experiments. We can now compare the behavior and performance of our LE-augmented model to its original archetype. Since large networks using the original model require prohibitively long training times when simulated with full dynamics rather than just their steady state [8], we use small networks and a small synthetic dataset as a benchmark (Fig. 3c).

Our microcircuit model can learn perfect classification even for very short presentation times. In contrast, the original model without the prospective mechanism stagnates at high error rates even for this simple task. As discussed earlier, this can be traced back to the learning process being disrupted during relaxation. Without prospective dynamics, the model requires presentation times on the order of $100\,\tau^{\text{eff}}$ to achieve perfect accuracy (Fig. 3d). In contrast, LE only degrades for presentation times below $0.1\,\tau^{\text{eff}}$, which is due to the limited resolution of our numerical integration method. Thus, incorporating LE into cortical microcircuits can bring the required presentation times into biologically plausible regimes, allowing networks to deal with rich sensory data.

## 6 Robustness to substrate imperfections

Computer simulations often assume perfectly homogeneous parameters across the network. Models can hence inadvertently rely on this homogeneity, resulting in unpredictable behavior and possibly fatal dysfunction when faced with the physics of analog substrates which are characterized by both heterogeneity in their components as well as temporal perturbation of their dynamics. Therefore, we consider robustness to spatio-temporal noise to represent a necessary prerequisite for any mathematical model aspiring to physical implementation, be it biological or artificial.

Spatial noise reflects the individuality of cortical neurons or the heterogeneity arising from device mismatch in hardware. Here, we focus on the heterogeneity of time constants; in contrast to, for example, variability in synaptic parameters or activation functions, these variations can not be "trained away" by adapting synaptic weights [36–38]. The two time constants that govern neuron dynamics in our model, namely integration (Eqn. 3) and prospective coding (Eqn. 1), previously assumed to be identical, are affected independently by such variability. To differentiate between the two, we assign the prospective dynamics their own time constant: $\breve{u}^{\text{r}} := u + \tau^{\text{r}}\dot{u}$. We can now model heterogeneity as independent, multiplicative Gaussian noise on all time constants: $\tau^{\text{m/r}} \rightarrow (1+\xi)\tau^{\text{m/r}}$, with $\xi \sim \mathcal{N}(0, \sigma_\tau^2)$; we use multiplicative noise to emphasize that our model is agnostic to absolute time scales, so only relative relationships between specific time constants matter.

Due to the resulting mismatches between the timing of neuronal input and output, neuronal outputs suffer from exponential transients, leading to relaxation issues similar to the ones we already addressed in detail. However, depending on the transmission of top-down signals, the effects on learning performance can be very different. According to the formal theory, backward errors use the correct prospective voltages: $e \propto [\breve{u}^{\text{m}} - W\varphi(\breve{u}^{\text{r}})]$ (Eqns. 4 and 5); this allows robust learning even for relatively large perturbations in the forward signals (Fig. 4a). In contrast, in biophysical implementations such as the microcircuits discussed above, neurons can only transmit a single output signal $\varphi(\breve{u}_i^{\text{r}})$, which consequently also affects the errors: $e \propto [\breve{u}^{\text{r}} - W\varphi(\breve{u}^{\text{r}})]$. Since deviations due

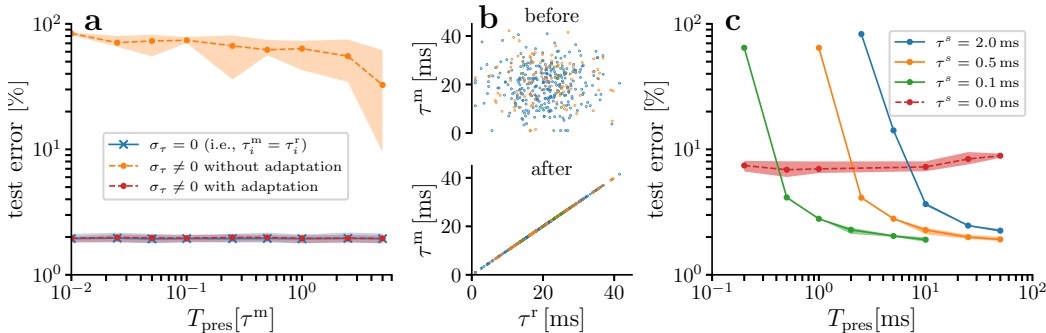

**Figure 4: Robustness of LE.** We show test errors on MNIST (min, max and mean over several seeds) after 100 training epochs for different input presentation times. **(a)** Effects of heterogeneity of time constants with (red) and without (orange) adaptation. Baseline with homogeneous time constants in blue for comparison. **(b)** Integration and prospective time constant pairs $(\tau^{\mathrm{m}}, \tau^{\mathrm{r}})$ for individual neurons within a network before (top) and after (bottom) development. Colors encode the layer to which the neurons belong. **(c)** Effects of temporal noise of width $\sigma_{\mathrm{r}} = 0.2\, r_{\mathrm{max}}$ with (solid) and without (dashed) synaptic filtering for different synaptic time constants.

to a mismatch between integration and prospective time constants persist on the time scale of $\tau^{\mathrm{m}}$ (see SI), even small amounts of mismatch can lead to a significant loss in performance (Fig. 4a).

Here we address this issue by introducing a neuron-local adaptation mechanism that corrects the difference between prospective voltages $\breve{u}^{\mathrm{m}}$ and $\breve{u}^{\mathrm{r}}$ induced by mismatching time constants:

$$\dot{\boldsymbol{\tau}}^{\mathrm{m}} = \eta_\tau \left( \breve{\boldsymbol{u}}^{\mathrm{r}} - \boldsymbol{W}\boldsymbol{r} - \boldsymbol{b} \right) \dot{\boldsymbol{u}} \,. \tag{8}$$

In biological substrates, this could, for example, correspond to an adaptation of the transmembrane ion channel densities, whereas on many neuromorphic substrates, neuronal time constants can be adapted individually [39]. Before training the network we allow it to go through a "developmental phase" in which individual neurons are not affected by top-down errors and learn to match their prospective firing to their membrane dynamics (Fig. 4b), thus recovering the performance of networks with perfectly matched time constants (Fig. 4a). This developmental phase merely consists of presenting input samples to the network, for a duration that depends on the required matching precision. Here, we achieved mismatches below 1‰ within the equivalent of 20 training epochs. Note that neuronal time constants remain diverse after this phase, but are matched in a way that enables fast reaction of downstream areas to sensory stimuli – an ability that is certainly helpful for survival. This aligns well with in-vivo observations of parameters that are highly diverse between neurons and individuals, but are fine-tuned within each neuron in order to reliably produce a desired input-output relationship [40].

We next model additive temporal noise on neuronal outputs as might be induced, for example, by noisy transmission channels: $r \rightarrow r + \xi$, with $\xi \sim \mathcal{N}(0, \sigma_{\mathrm{r}}^2)$. Formally, this turns membrane dynamics into a Langevin process. These perturbations add up over consecutive layers and can also accumulate over time due to recurrent interactions in the network. This introduces noise to the weight updates that can impair learning performance. Due to their slow integration of inputs, traditional neuron models filter out this noise, but our prospective mechanism effectively removes this filter. We thus need an additional denoising mechanism, for which we again turn to biology: by introducing synaptic filtering with a short time constant $\tau^{\mathrm{s}}$, synaptic input is denoised before reaching the membrane; formally, $r = \varphi(\breve{u}^{\mathrm{m}})$ is replaced by a low-pass-filtered $\bar{r}^{\mathrm{s}}$ in the energy function (Eqn. 2, see also SI). This narrow filter also mitigates possible numerical instabilities, as discussed in Section 3.

Networks equipped with synaptic filters learn reliably even in the presence of significant noise levels (Fig. 4c). However, introducing this filter affects the quasi-instantaneity of computation in our networks, which then require longer input presentation times. Even so, these presentation times need only be "long" with respect to the characteristic time constant of relaxation mismatches – in this case, $\tau^{\mathrm{s}}$. Thus, for the typical scenario of white noise described above, minuscule $\tau^{\mathrm{s}}$ on and even below biological time scales (see, e.g., [41–43]) can achieve effective denoising, without significantly affecting the advantages conferred by prospective coding. In conclusion, the demonstrated robustness of our model to spatial and temporal substrate imperfections introduced by simple, biologically inspired mechanisms, make it a promising candidate for implementation in analog physical systems.

## 7 Implications and limitations

In this section, we briefly address several interesting questions that arise from physical embeddings of LE, both in vivo and in silico.

Within the context of LE, the microcircuit model used to implement error backpropagation carries several implications for cortical phenomenology beyond specific connectivity patterns. In particular, we expect that during active, attentive sensory processing, both during and after learning, cortical pyramidal cells and inhibitory interneurons will react simultaneously to changes in the sensory stimulus. Furthermore, we would also expect such synchronization at the level of cross-cortical networks, for example across the ventral visual stream. This should be observable in large-scale activity recordings with sufficient temporal resolution, for example in iEEG data.

A further implication of our framework is that plasticity is, in principle, equanimous about the speed of sensory input changes. In particular, learning should be possible without neurons ever reaching a steady state. We would argue that the ability of mammals to learn from a continuously, and often quickly changing input stream already provides evidence for such quick processing and learning. In fact, stimulus presentation times for humans in a subliminal face-word association paradigm can be as short as 17 ms and still induce association learning [44]. Furthermore, we expect in-vivo plasticity in cortical subnetworks responsible for such forms of pattern recognition to explicitly depend on prospective neuronal states; this would contrast with other paradigms that only propose a dependence on instantaneous (e.g., [35]) or even low-pass-filtered (e.g., [45]) state variables.

By explicitly approximating error backpropagation, our framework inherits its challenges with respect to biological plausibility, in particular the weight transport problem. In part, this is addressed by feedback alignment, as used in our cortical microcircuits. However, these weights can themselves be plastic in order to further boost learning performance [46–49]. The representation of errors by nudging neuronal activity also has the effect of diluting error signals (in addition to the vanishing gradient problem); this could be mitigated by adapting learning rates as a function of layer identity.

One major aspect of biology that our framework does not explicitly address is the spike-based communication between neurons. In silico, this represents less of an obstacle, because pulsed communication packets can easily carry more information than just the pulse arrival times by including additional payload. In vivo, a similar role could be played by inter-spike-intervals within spike doublets or bursts, or by the precise spike timing used in, for example, spike latency codes [50].

With respect to neuromorphic implementation, we should stress that our robustness analysis only represents a starting point and not a quantitatively faithful study of the expected effects in analog/mixed-signal hardware. While the investigated forms of spatiotemporal noise certainly are among the most salient, there are many other substrate-induced distortive effects to be considered, such as neuronal delays, limited bandwidth or limited synaptic real-estate [51, 52]. Ultimately, an actual demonstration in silico will have to be the definitive arbiter of the neuromorphic feasibility of LE.

## 8 Conclusion

We have introduced a new framework for inference and learning in physical systems composed of computational elements with finite response times. Our model rests on four simple axioms: prospective coding (Eqn. 1), neuronal mismatch energy (Eqn. 2), energy conservation under neuronal dynamics (Eqn. 3) and gradient descent on the energy under synaptic plasticity (Eqn. 5). In particular, incorporating the simple, biologically inspired mechanism of prospective coding allows us to avoid critical issues and scalability bottlenecks inherent to many current models of approximate BP in cortex. Furthermore, we have demonstrated robustness of the resulting implementations to substrate imperfections, a prerequisite for deployment in analog neuronal systems, both biological and artificial.

Our framework carries implications both for neuroscience and for the design of neuromorphic hardware. The prospective mechanism described here would allow biological circuits to respond much faster than previously assumed. Furthermore, our framework suggests that both inference and learning take place on prospective, rather than instantaneous neuronal quantities. From a hardware perspective, this lifts the previously perceived limitations of slow analog components (as compared to digital ones) without relinquishing their power efficiency.

# 9 Acknowledgements

This work has received funding from the European Union 7th Framework Programme under grant agreement 604102 (HBP), the Horizon 2020 Framework Programme under grant agreements 720270, 785907 and 945539 (HBP), the Swiss National Science Foundation (SNSF, Sinergia grant CRSII5-180316) and the Manfred Stärk Foundation. We acknowledge the use of Fenix Infrastructure resources, which are partially funded from the European Union's Horizon2020 research and innovation programme through the ICEI project under the grant agreement No. 800858. Furthermore, we thank Mathieu Le Douairon, Reinhard Dietrich and the Insel Data Science Center for the usage and outstanding support of their Research HPC Cluster. A very special thank you goes to Ellis, for his uniquely wholesome influence during the final stretch of this work.

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
