## Supplementary Information

## 10 Relation between low-pass filter and lookahead

In general, the prospective (or lookahead) voltage $\breve{u}^{\mathrm{r}}$ that enters the activation function "looks into the future" with a time horizon of $\tau^{\mathrm{r}}$, that is in general different from $\tau^{\mathrm{m}}$ (cf. Eqn. 1):

$$\breve{u}^{\mathrm{r}}(t) := \left(1 + \tau^{\mathrm{r}}\frac{\mathrm{d}}{\mathrm{d}t}\right) u(t) . \tag{9}$$

On the other hand, the exponential low-pass filter (LPF) with time constant $\tau^{\mathrm{x}}$ of some time-dependent quantity $u(t)$ is given by the average over the past, weighted with an exponential kernel

$$\overline{u}^{\mathrm{x}}(t) := \frac{1}{\tau^{\mathrm{x}}} \int_{-\infty}^{t} u(t') \mathrm{e}^{(t'-t)/\tau^{\mathrm{x}}} \, \mathrm{d}t' . \tag{10}$$

Usually, $\tau^{\mathrm{x}}$ is either the membrane or the synaptic time constant. To see what happens for a certain neuron in the case of heterogeneous time constants, $(\tau^{\mathrm{m}} \neq \tau^{\mathrm{r}})$, we can compute

$$\frac{\mathrm{d}}{\mathrm{d}t}\overline{u}^{\mathrm{m}} = \frac{\mathrm{d}}{\mathrm{d}t}\frac{1}{\tau^{\mathrm{m}}} \int_{-\infty}^{t} u(t') \mathrm{e}^{(t'-t)/\tau^{\mathrm{m}}} \, \mathrm{d}t' \tag{11}$$

$$= \frac{1}{\tau^{\mathrm{m}}} \left( \underbrace{u(t') \mathrm{e}^{(t'-t)/\tau^{\mathrm{m}}}\Big|_{t'=t}}_{u(t)} + \int_{-\infty}^{t} u(t') \frac{\partial}{\partial t} \mathrm{e}^{(t'-t)/\tau^{\mathrm{m}}} \, \mathrm{d}t' \right) \tag{12}$$

$$= \frac{1}{\tau^{\mathrm{m}}} \left( u(t) - \underbrace{\frac{1}{\tau^{\mathrm{m}}} \int_{-\infty}^{t} u(t') \mathrm{e}^{(t'-t)/\tau^{\mathrm{m}}} \, \mathrm{d}t'}_{\overline{u}^{\mathrm{m}}(t)} \right) \tag{13}$$

$$= \frac{1}{\tau^{\mathrm{m}}} \left( u(t) - \overline{u}^{\mathrm{m}}(t) \right) \tag{14}$$

to obtain the general expression

$$\breve{\overline{u}}^{\mathrm{r}}(t) = \left(1 + \tau^{\mathrm{r}}\frac{\mathrm{d}}{\mathrm{d}t}\right)\overline{u}^{\mathrm{m}}(t) = \overline{u}^{\mathrm{m}} + \frac{\tau^{\mathrm{r}}}{\tau^{\mathrm{m}}}\left(u(t) - \overline{u}^{\mathrm{m}}(t)\right) \tag{15}$$

$$= \frac{\tau^{\mathrm{r}}}{\tau^{\mathrm{m}}}u(t) + \frac{\tau^{\mathrm{m}} - \tau^{\mathrm{r}}}{\tau^{\mathrm{m}}}\overline{u}^{\mathrm{m}}(t) . \tag{16}$$

As mentioned in Section 6, different prospective and membrane time constants lead to deviations that persist on the time scale of $\tau^{\mathrm{m}}$ since the second term in Eqn. 15 is still proportional to the low-pass filtered voltage $\overline{u}^{\mathrm{m}}$, which relaxes with the specific time constant $\tau^{\mathrm{m}}$. On the other hand, in case of equal time constants $\tau^{\mathrm{m}} = \tau^{\mathrm{r}}$ it immediately follows that lookahead and LPF are inverse operations:

$$\breve{\overline{u}}(t) = \left(1 + \tau\frac{\mathrm{d}}{\mathrm{d}t}\right)\overline{u}(t) = \overline{u}(t) + \tau\dot{\overline{u}}(t) = u(t) . \tag{17}$$

## 11 Detailed derivation of the neuronal dynamics

Eqn. 3 represents the solution for a stationary energy with respect to the prospective voltage $\breve{u}^{\mathrm{m}}$. In the following, we show the detailed derivation for an arbitrary component $i$:

$$\frac{\partial E}{\partial \breve{u}_i^{\mathrm{m}}} = \frac{\partial}{\partial \breve{u}_i^{\mathrm{m}}}\frac{1}{2}\sum_{j,k}\left\| \breve{u}_j^{\mathrm{m}} - W_{jk}\varphi(\breve{u}_k^{\mathrm{m}}) \right\|^2 \tag{18}$$

$$= \sum_{j,k,l}\left[ \breve{u}_j^{\mathrm{m}} - W_{jk}\varphi(\breve{u}_k^{\mathrm{m}}) \right]\frac{\partial}{\partial \breve{u}_i}\left[ \breve{u}_j^{\mathrm{m}} - W_{jl}\varphi(\breve{u}_l^{\mathrm{m}}) \right] \tag{19}$$

$$= \sum_{j,k,l}\left[ \breve{u}_j^{\mathrm{m}} - W_{jk}\varphi(\breve{u}_k^{\mathrm{m}}) \right]\left[ \delta_{ij} - \delta_{il}W_{jl}\varphi'(\breve{u}_l^{\mathrm{m}}) \right] \tag{20}$$

$$= \breve{u}_i^{\mathrm{m}} - \sum_{k}W_{ik}\varphi(\breve{u}_k^{\mathrm{m}}) - \varphi'(\breve{u}_i^{\mathrm{m}})\sum_{j,k}W_{ij}^{\mathrm{T}}\left[ \breve{u}_j^{\mathrm{m}} - W_{jk}\varphi(\breve{u}_k^{\mathrm{m}}) \right] \tag{21}$$

and therefore

$$0 = \frac{\partial E}{\partial \breve{u}_i^{\mathrm{m}}} \implies \tau^{\mathrm{m}} \dot{u}_i = -u_i + \sum_k W_{ik} \varphi(\breve{u}_k^{\mathrm{m}}) + \varphi'(\breve{u}_i^{\mathrm{m}}) \sum_{j,k} W_{ij}^{\mathrm{T}} \left[ \breve{u}_j^{\mathrm{m}} - W_{jk} \varphi(\breve{u}_k^{\mathrm{m}}) \right] \quad (22)$$

## 11.1 Neuronal dynamics with synaptic filtering

To include synaptic filtering in our theory, we introduce an additional LPF as in Eqn. 10 with time constant $\tau^{\mathrm{s}}$. For this, it is sufficient to replace firing rates with filtered rates, $\varphi \to \overline{\varphi}^{\mathrm{s}}$, in the total energy $E$:

$$E(\breve{u}^{\mathrm{m}}) := \frac{1}{2} \left\| \breve{u}^{\mathrm{m}} - W \overline{\varphi}^{\mathrm{s}}(\breve{u}^{\mathrm{m}}) - b \right\|^2 + \beta \mathcal{L}(\breve{u}^{\mathrm{m}}) \, . \quad (23)$$

Deriving the neuronal dynamics from a vanishing gradient $\nabla_{\breve{u}^{\mathrm{m}}} E$ as before now yields

$$\tau^{\mathrm{m}} \dot{u} = -u + W \overline{\varphi}^{\mathrm{s}}(\breve{u}^{\mathrm{m}}) + b + e \, , \quad (24)$$

where the error term now reads

$$e = \overline{\varphi}^{\mathrm{s}}(\breve{u}^{\mathrm{m}}) W^{\mathrm{T}} \left[ \breve{u}^{\mathrm{m}} - W \overline{\varphi}^{\mathrm{s}}(\breve{u}^{\mathrm{m}}) - b \right] \, . \quad (25)$$

These are essentially the same equations as before, just with the rates replaced with their filtered version.

# 12 General formulation for arbitrary connectivity functions

Here we consider a generalization of the energy function from the main manuscript that includes arbitrary "connectivity functions" $f$ with parameters $\theta$:

$$E(\breve{u}^{\mathrm{m}}) := \frac{1}{2} \left\| \breve{u}^{\mathrm{m}} - f(\varphi(\breve{u}^{\mathrm{m}}), \theta) \right\|^2 + \beta \mathcal{L}(\breve{u}^{\mathrm{m}}) \, . \quad (26)$$

Again, we derive neuron dynamics by requiring $\nabla_{\breve{u}^{\mathrm{m}}} E(\breve{u}^{\mathrm{m}}) = 0$. For simplicity, we compute it element wise, shown here for a neuron that does not directly contribute to the loss $\mathcal{L}$:

$$\begin{aligned}
\frac{\partial E(\breve{u}^{\mathrm{m}})}{\partial \breve{u}_i^{\mathrm{m}}} &= \breve{u}_i^{\mathrm{m}} - f_i(\varphi(\breve{u}^{\mathrm{m}}), \theta) + \sum_j \frac{\partial}{\partial \breve{u}_i^{\mathrm{m}}} \frac{1}{2} \left( \breve{u}_j^{\mathrm{m}} - f_j(\varphi(\breve{u}^{\mathrm{m}}), \theta) \right)^2 \\
&= \breve{u}_i^{\mathrm{m}} - f_i(\varphi(\breve{u}^{\mathrm{m}}), \theta) - \sum_j \frac{\partial \varphi_i}{\partial \breve{u}_i^{\mathrm{m}}} \frac{\partial f_j(\varphi(\breve{u}^{\mathrm{m}}), \theta)}{\partial \varphi_i} \left( \breve{u}_j^{\mathrm{m}} - f_j(\varphi(\breve{u}^{\mathrm{m}}), \theta) \right) \quad (27)
\end{aligned}$$

From this we can, for example, obtain the neuron dynamics described in the main manuscript by choosing a linear connectivity function $f_i = \sum_j w_{ij} \varphi(\breve{u}_j^{\mathrm{m}}) + b_i$.

# 13 Simulation details

## 13.1 Gradient-based model

For Fig. 1c we consider the neuron dynamics from the main manuscript, but replace prospective membrane potentials with their instantaneous version. Furthermore, we consider a squared loss with some fixed target $t^*$. To avoid artificially introducing an additional mismatch problem, we add an exponential low-pass filter to the error terms; this prevents the model to reduce weights to zero in the absence of a teacher due to a continuous mismatch between instantaneous bottom-up predictions and slow neuronal responses. This results in the following dynamics for the two neurons:

$$\tau^{\mathrm{m}} \dot{u}_1 = -u_1 + w_1 r_{\mathrm{in}} + \varphi'(u_1) w_2 (u_2 - w_2 \overline{\varphi}(u_1)) \quad (28)$$

$$\tau^{\mathrm{m}} \dot{u}_2 = -u_2 + w_2 \varphi(u_1) + \beta(t^* - u_2) \quad (29)$$

As in the main manuscript, plasticity is defined as stochastic gradient descent on the energy; as above, we consider low-pass filtered variants of inputs:

$$\Delta w_i = \eta (u_i - W \overline{r}_{i-1}) \overline{r}_{i-1} \quad (30)$$

Here we use linear activation functions and the following parameters: $\mathrm{d}t = 0.001, \tau^{\mathrm{m}} = 10\mathrm{ms}, \beta \in \{0, 0.9\}, \eta_W = 0.0005$.

## 13.2 Numerical implementation

In order to carry out our simulations we had to discretize the differential equations, which we state here again for clarity:

$$\tau^{\mathrm{m}}\dot{\boldsymbol{u}}_\ell(t) = -\boldsymbol{u}_\ell(t) + \boldsymbol{W}_\ell\boldsymbol{\varphi}(\breve{\boldsymbol{u}}_\ell^{\mathrm{m}}(t)) + \boldsymbol{\varphi}'(\breve{\boldsymbol{u}}_\ell^{\mathrm{m}}(t))\boldsymbol{W}_{\ell+1}^{\mathrm{T}}\left[\breve{\boldsymbol{u}}_{\ell+1}^{\mathrm{m}}(t) - \boldsymbol{W}_{\ell+1}\boldsymbol{\varphi}(\breve{\boldsymbol{u}}_\ell^{\mathrm{m}}(t))\right] \tag{31}$$

$$\dot{\boldsymbol{W}}_\ell(t) = \eta\left[\breve{\boldsymbol{u}}_\ell^{\mathrm{m}}(t) - \boldsymbol{W}_\ell\boldsymbol{\varphi}(\breve{\boldsymbol{u}}_{\ell-1}^{\mathrm{m}}(t))\right]\boldsymbol{\varphi}(\breve{\boldsymbol{u}}_{\ell-1}^{\mathrm{m}}(t)) . \tag{32}$$

First, observe that $\dot{\boldsymbol{u}}$ appears on both sides of the first equation: explicitly on the left hand side, and implicitly on the right, in the definition of $\breve{\boldsymbol{u}}^{\mathrm{m}}(t) = \boldsymbol{u}(t) + \tau^{\mathrm{m}}\dot{\boldsymbol{u}}(t)$. To resolve this circular dependency, we define $\breve{\boldsymbol{u}}^{\mathrm{m}}(t + \mathrm{d}t) = \boldsymbol{u}(t) + \tau^{\mathrm{m}}\dot{\boldsymbol{u}}(t)$, which works well for small enough $\mathrm{d}t$.

We first consider the neuronal update and use forward Euler to rewrite the derivative as a finite difference with time step $\mathrm{d}t$

$$\tau^{\mathrm{m}}\frac{\boldsymbol{u}_\ell(t + \mathrm{d}t) - \boldsymbol{u}_\ell(t)}{\mathrm{d}t} = -\boldsymbol{u}_\ell(t) + \boldsymbol{W}_\ell\boldsymbol{\varphi}(\breve{\boldsymbol{u}}_\ell^{\mathrm{m}}(t)) + \boldsymbol{e}_\ell(t) \tag{33}$$

with

$$\boldsymbol{e}_\ell(t) := \boldsymbol{\varphi}'(\breve{\boldsymbol{u}}_\ell^{\mathrm{m}}(t))\boldsymbol{W}_{\ell+1}^{\mathrm{T}}\left[\breve{\boldsymbol{u}}_{\ell+1}^{\mathrm{m}}(t) - \boldsymbol{W}_{\ell+1}\boldsymbol{\varphi}(\breve{\boldsymbol{u}}_\ell^{\mathrm{m}}(t))\right] \tag{34}$$

and solve for $\boldsymbol{u}_\ell(t + \mathrm{d}t)$ to obtain

$$\boldsymbol{u}_\ell(t + \mathrm{d}t) = \boldsymbol{u}_\ell(t) + \mathrm{d}t\boldsymbol{\Delta}\boldsymbol{u}_\ell(t) , \tag{35}$$

where

$$\boldsymbol{\Delta}\boldsymbol{u}_\ell(t) = \frac{1}{\tau^{\mathrm{m}}}\left[-\boldsymbol{u}_\ell(t) + \boldsymbol{W}_\ell\boldsymbol{\varphi}(\breve{\boldsymbol{u}}_\ell^{\mathrm{m}}(t)) + \boldsymbol{e}_\ell(t)\right] . \tag{36}$$

Similarly, we use forward Euler for weight dynamics:

$$\frac{\boldsymbol{W}_\ell(t + \mathrm{d}t) - \boldsymbol{W}_\ell(t)}{\mathrm{d}t} = \eta\left[\breve{\boldsymbol{u}}_\ell^{\mathrm{m}}(t) - \boldsymbol{W}_\ell\boldsymbol{\varphi}(\breve{\boldsymbol{u}}_{\ell-1}^{\mathrm{m}}(t))\right]\boldsymbol{\varphi}(\breve{\boldsymbol{u}}_{\ell-1}^{\mathrm{m}}(t)) \tag{37}$$

and

$$\boldsymbol{W}_\ell(t + \mathrm{d}t) = \boldsymbol{W}_\ell(t) + \mathrm{d}t\boldsymbol{\Delta}\boldsymbol{W}_\ell(t) , \tag{38}$$

with

$$\boldsymbol{\Delta}\boldsymbol{W}_\ell(t) = \eta\left[\breve{\boldsymbol{u}}_\ell^{\mathrm{m}}(t) - \boldsymbol{W}_\ell\boldsymbol{\varphi}(\breve{\boldsymbol{u}}_{\ell-1}^{\mathrm{m}}(t))\right]\boldsymbol{\varphi}(\breve{\boldsymbol{u}}_{\ell-1}^{\mathrm{m}}(t)) \tag{39}$$

For both $\boldsymbol{e}_\ell(t)$ and $\boldsymbol{\Delta}\boldsymbol{W}_\ell(t)$ it is crucial to combine the $\breve{\boldsymbol{u}}^{\mathrm{m}}(t)$ with the input $\boldsymbol{W}_\ell\boldsymbol{\varphi}(\breve{\boldsymbol{u}}_{\ell-1}^{\mathrm{m}}(t))$ that was also used in in computing $\breve{\boldsymbol{u}}^{\mathrm{m}}(t)$. Pseudo-code for our vanilla implementation can be found in Algorithm 1.

---

**Algorithm 1** Pseudo-code for the multi-layer implementation of Latent Equilibrium (LE)

---

1: **for all** layers $\ell$ from 1 (input) to $N$ (output) **do**
2: $\quad\boldsymbol{e}_\ell(t) \leftarrow (1 - \delta_{\ell N})\boldsymbol{\varphi}'(\breve{\boldsymbol{u}}_\ell^{\mathrm{m}}(t))\boldsymbol{W}_{\ell+1}^{\mathrm{T}}(t)\left[\breve{\boldsymbol{u}}_{\ell+1}^{\mathrm{m}}(t) - \boldsymbol{W}_{\ell+1}(t)\boldsymbol{\varphi}(\breve{\boldsymbol{u}}_\ell^{\mathrm{m}}(t))\right] + \delta_{\ell N}\boldsymbol{e}^{\mathrm{trg}}(t)$
3: $\quad\boldsymbol{\Delta}\boldsymbol{u}_\ell(t) \leftarrow \tau^{-1}\left[-\boldsymbol{u}_\ell(t) + \boldsymbol{W}_\ell(t)\boldsymbol{r}_{\ell-1}(t) + \boldsymbol{e}_\ell(t)\right]$
4: $\quad\boldsymbol{u}_\ell(t + \mathrm{d}t) \leftarrow \boldsymbol{u}_\ell(t) + \mathrm{d}t\,\boldsymbol{\Delta}\boldsymbol{u}_\ell(t)$
5: $\quad\breve{\boldsymbol{u}}_\ell(t + \mathrm{d}t) \leftarrow \boldsymbol{u}_\ell(t) + \tau\boldsymbol{\Delta}\boldsymbol{u}_\ell(t)$
6: $\quad$**if** synaptic plasticity **then**
7: $\quad\quad\boldsymbol{\Delta}\boldsymbol{W}_\ell(t) \leftarrow \eta\,\boldsymbol{e}_\ell(t) \cdot \boldsymbol{\varphi}(\breve{\boldsymbol{u}}_{\ell-1}^{\mathrm{m}}(t))^{\mathrm{T}}$
8: $\quad\quad\boldsymbol{W}_\ell(t + \mathrm{d}t) \leftarrow \boldsymbol{W}_\ell(t) + \mathrm{d}t\,\boldsymbol{\Delta}\boldsymbol{W}_\ell(t)$
9: $\quad$**end if**
10: **end for**

---

## 14 Microcircuit details

The somatic membrane potential of hidden layer pyramidal cells, interneurons, and top-layer pyramidal cells is described by the following differential equations:

$$C_{\mathrm{m}}\dot{\boldsymbol{u}}_\ell^{\mathrm{P}} = g_{\mathrm{l}}\left(E_{\mathrm{l}} - \boldsymbol{u}_\ell^{\mathrm{P}}\right) + g^{\mathrm{bas}}\left(\boldsymbol{v}_\ell^{\mathrm{bas}} - \boldsymbol{u}_\ell^{\mathrm{P}}\right) + g^{\mathrm{api}}\left(\boldsymbol{v}_\ell^{\mathrm{api}} - \boldsymbol{u}_\ell^{\mathrm{P}}\right) , \tag{40}$$

$$C_{\mathrm{m}}\dot{\boldsymbol{u}}_\ell^{\mathrm{I}} = g_{\mathrm{l}}\left(E_{\mathrm{l}} - \boldsymbol{u}_\ell^{\mathrm{I}}\right) + g^{\mathrm{den}}\left(\boldsymbol{v}_\ell^{\mathrm{den}} - \boldsymbol{u}_\ell^{\mathrm{I}}\right) + i^{\mathrm{nudge, I}} , \tag{41}$$

$$C_{\mathrm{m}}\dot{\boldsymbol{u}}_N^{\mathrm{P}} = g_{\mathrm{l}}\left(E_{\mathrm{l}} - \boldsymbol{u}_N^{\mathrm{P}}\right) + g^{\mathrm{bas}}\left(\boldsymbol{v}_N^{\mathrm{bas}} - \boldsymbol{u}_N^{\mathrm{P}}\right) + i^{\mathrm{nudge, tgt}} . \tag{42}$$

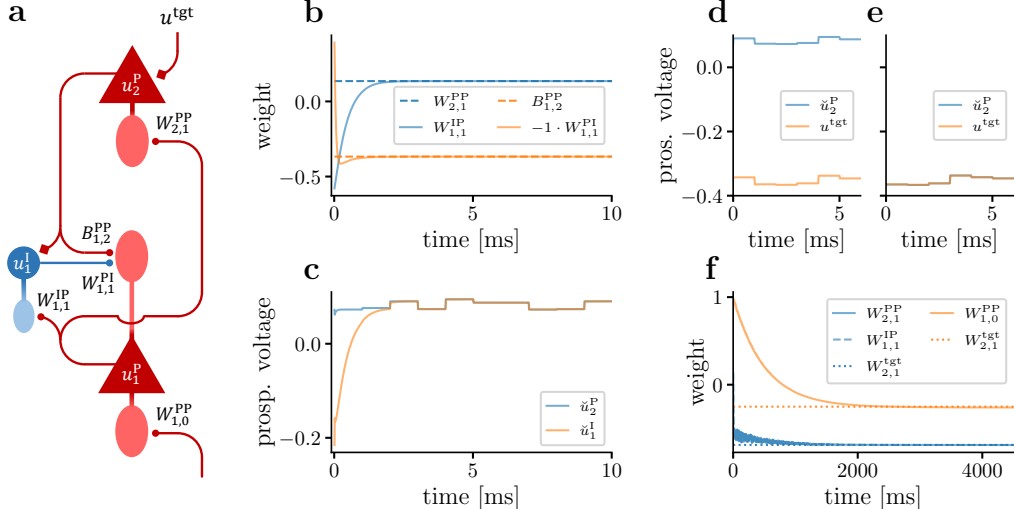

**Figure 5: Learning to mimic a teacher microcircuit with LE.** **(a)** Microcircuit architecture following [1]. **(b)** Learning of the lateral weights $W_{1,1}^{\text{IP}}$ and $W_{1,1}^{\text{PI}}$ to implement the self-predicting state. **(c)** Prospective membrane voltages during learning of the self-predicting state where (in absence of a target) the top-down activity is matched by the activity of the interneuron. **(d, e)** Comparison between the prospective membrane voltage $\breve{u}_2^{\text{P}}$ of the output pyramidal neuron and the target voltage $u^{\text{tgt}}$ before (d) and after (e) training. **(f)** Weight evolution during learning.

Target signals to interneurons and top-layer pyramidal neurons are modeled as a conductance-based input to the respective somatic compartments:

$$i^{\text{nudge, I}} = g^{\text{nudge, I}} \left( u_{\ell+1}^{\text{P}} - u_\ell^{\text{I}} \right) \ , \tag{43}$$

$$i^{\text{nudge, tgt}} = g^{\text{nudge, tgt}} \left( u^{\text{tgt}} - u_N^{\text{P}} \right) \ . \tag{44}$$

The target signal for the top-layer pyramidal is determined by the training set. For the interneurons, the somatic membrane potentials of the pyramidal neurons in the layer above serve as targets. The membrane potentials of the dendritic compartments instantaneously follow their inputs:

$$\boldsymbol{v}_\ell^{\text{bas}} = \boldsymbol{W}_{\ell,\ell-1}^{\text{PP}} \varphi \left( \boldsymbol{u}_{\ell-1}^{\text{P}} \right) \ , \tag{45}$$

$$\boldsymbol{v}_\ell^{\text{api}} = \boldsymbol{B}_{\ell,\ell+1}^{\text{PP}} \varphi \left( \boldsymbol{u}_{\ell+1}^{\text{P}} \right) + \boldsymbol{W}_{\ell,\ell}^{\text{PI}} \varphi \left( \boldsymbol{u}_\ell^{\text{I}} \right) \ , \tag{46}$$

$$\boldsymbol{v}_\ell^{\text{den}} = \boldsymbol{W}_{\ell,\ell}^{\text{IP}} \varphi \left( \boldsymbol{u}_\ell^{\text{P}} \right) \ . \tag{47}$$

All synapses except the top-down connections are plastic. The learning rules are described by

$$\dot{\boldsymbol{W}}_{\ell,\ell-1}^{\text{PP}} = \eta_\ell^{\text{PP}} \left[ \varphi \left( \boldsymbol{u}_\ell^{\text{P}} \right) - \varphi \left( \frac{g^{\text{bas}}}{g_{\text{l}} + g^{\text{bas}} + g^{\text{api}}} \boldsymbol{v}_\ell^{\text{bas}} \right) \right] \varphi \left( \boldsymbol{u}_{\ell-1}^{\text{P}} \right) \ , \tag{48}$$

$$\dot{\boldsymbol{W}}_{\ell,\ell}^{\text{IP}} = \eta_\ell^{\text{IP}} \left[ \varphi \left( \boldsymbol{u}_\ell^{\text{I}} \right) - \varphi \left( \frac{g^{\text{den}}}{g_{\text{l}} + g^{\text{den}}} \boldsymbol{v}_\ell^{\text{den}} \right) \right] \varphi \left( \boldsymbol{u}_\ell^{\text{P}} \right) \ , \tag{49}$$

$$\dot{\boldsymbol{W}}_{\ell,\ell}^{\text{PI}} = \eta_\ell^{\text{PI}} \left[ -\boldsymbol{v}_\ell^{\text{api}} \right] \varphi \left( \boldsymbol{u}_\ell^{\text{I}} \right) \ . \tag{50}$$

Here, the weights of the top-down connections $B^{\text{PP}}$ are static and random.

The differential equations for the somatic membrane potentials of all neuron types can be rewritten in a simpler form which also increases their numerical stability:

$$C_{\text{m}} \dot{\boldsymbol{u}} = \frac{1}{\tau_{\text{eff}}} \left( \boldsymbol{u}^{\text{eff}} - \boldsymbol{u} \right) \ , \tag{51}$$

$$\tau_{\text{eff}} = \frac{C_{\text{m}}}{g_{\text{l}} + g^{\text{bas/den}} + g^{\text{api/nudge}}} \ . \tag{52}$$

Here, $\boldsymbol{u}^{\text{eff}}$ is the effective reversal potential defined as:

$$\boldsymbol{u}_\ell^{\text{eff,P}} = \frac{g_\text{l} E_\text{l} + g^{\text{bas}} \boldsymbol{v}_\ell^{\text{bas}} + g^{\text{api}} \boldsymbol{v}_\ell^{\text{api}}}{g_\text{l} + g^{\text{bas}} + g^{\text{api}}} \;, \tag{53}$$

$$\boldsymbol{u}_\ell^{\text{eff,I}} = \frac{g_\text{l} E_\text{l} + g^{\text{den}} \boldsymbol{v}_\ell^{\text{den}} + g^{\text{nudge, I}} \boldsymbol{u}_{\ell+1}^{\text{P}}}{g_\text{l} + g^{\text{den}} + g^{\text{nudge, I}}} \;, \tag{54}$$

$$\boldsymbol{u}_N^{\text{eff,P}} = \frac{g_\text{l} E_\text{l} + g^{\text{bas}} \boldsymbol{v}_N^{\text{bas}} + g^{\text{nudge, tgt}} \boldsymbol{u}^{\text{tgt}}}{g_\text{l} + g^{\text{bas}} + g^{\text{nudge, tgt}}} \tag{55}$$

if a target is provided. If no target is provided to the top-layer pyramidal neurons we assume $\boldsymbol{u}^{\text{tgt}} = \boldsymbol{u}_N^{\text{eff,P}}$ and the above equation simplifies to

$$\boldsymbol{u}_N^{\text{eff,P}} = \frac{g_\text{l} E_\text{l} + g^{\text{bas}} \boldsymbol{v}_N^{\text{bas}}}{g_\text{l} + g^{\text{bas}}} \;. \tag{56}$$

We include LE in the dendritic microcircuit by two simple modifications. First, the output rate of the neurons must depend on the prospective voltage: $\varphi(\boldsymbol{u}) \to \varphi(\breve{\boldsymbol{u}})$. Note that this includes also the rates in the calculation of dendritic membrane potentials (Eqns. 45 to 47) as well as the plasticity rules (Eqns. 48 to 50). Secondly, the nudging for the interneurons must depend on the prospective voltage of the pyramidal neurons above:

$$\boldsymbol{u}_\ell^{\text{eff,I}} = \frac{g_\text{l} E_\text{l} + g^{\text{den}} \boldsymbol{v}_\ell^{\text{den}} + g^{\text{nudge}} \breve{\boldsymbol{u}}_{\ell+1}^{\text{P}}}{g_\text{l} + g^{\text{den}} + g^{\text{nudge, I}}} \;. \tag{57}$$

For the simulation of the cortical networks shown in Fig. 3 and Fig. 5, we use the Euler integration method. Similarly to the LE networks without microcircuit connectivity, we break the circular dependency of $\breve{u}^{\text{m}}$ in an Euler integration step by defining $\breve{u}^{\text{m}}$ as a function of previous time steps (see Section 13.2).

Learning is split into two stages: first, the learning of the so-called self-predicting state and afterwards the learning of the actual task. The self-predicting state describes a configuration of weights in which, in the absence of target signals provided to the last layer, apical dendrites are always at rest and the somatic membrane potentials of the interneurons match the membrane potentials of the pyramidal neurons in the layer above. In this state, the network is able to correctly transport errors induced by the target signal to the apical compartments of the lower layer neurons.

Here we demonstrate, for a single microcircuit, the learning of the self-predicting state from a random initialization of weights, by presenting the network with random inputs, no targets to the output layer and using the learning rules given above with $\eta^{\text{PP}} = 0$ and $\eta^{\text{PI/IP}} \neq 0$ (Fig. 5 b, c). After the self-predicting state is learned, the network is taught to reproduce the input-output relationship produced by a teacher network (Fig. 5 d-f). For the learning of the task we set the learning rates to $\eta^{\text{PI}} = 0$ and $\eta^{\text{PP/IP}} \neq 0$. In the main manuscript (Fig. 3) we initialize the weights with

$$\boldsymbol{W}_{1,1}^{\text{IP}} = \frac{g^{\text{bas}}\left(g_\text{l} + g^{\text{den}}\right)}{g^{\text{den}}\left(g_\text{l} + g^{\text{bas}}\right)} \boldsymbol{W}_{2,1}^{\text{PP}} \quad \text{and} \tag{58}$$

$$\boldsymbol{W}_{1,1}^{\text{PI}} = -\boldsymbol{B}_{1,2}^{\text{PP}} \;, \tag{59}$$

thereby skipping the first learning stage and initializing the network directly in the self-predicting state. The full set of parameters used in Fig. 3 and Fig. 5 can be found in Section 15.2.

## 15  Parameters

### 15.1  Parameters used for classification experiments shown in Fig. 2

Table 1 lists all the parameters we used for the experiments shown in Fig. 2. This includes HIGGS and MNIST experiments with fully connected (FC) architectures in Fig. 2b and c as well as MNIST and CIFAR-10 experiments employing convolutional networks (ConvNets).

Standard artificial neural networks (ANNs) were trained with classical backpropagation (BP) using the same network topologies but with cross-entropy (CE) loss instead of the mean squared error (MSE) loss used for the LE experiments.

**Table 1:** Neuron, network and training parameters used to produce the results shown in Fig. 2.

| Symbol | Parameter name | Fig. 2a | Fig. 2b | Fig. 2c | Fig. 2d | Fig. 2e |
|---|---|---|---|---|---|---|
| **Neuron parameters** | | | | | | |
| $\tau^{\mathrm{m}}$ [ms] | membrane time constant | 10 | 20 | 10 | 10 | 10 |
| $\tau^{\mathrm{r}}$ [ms] | prospective time constant | 10 | 20 | 10 | 10 | 10 |
| $\tau^{\mathrm{s}}$ [ms] | synaptic time constant | 0 | 0 | 0 | 0 | 0 |
| $\varphi_\ell$ | activation | tanh | | hard sigmoid[1] | | |
| $\varphi_N$ | output activation | | | linear | | |
| **Network parameters** | | | | | | |
| | architecture | FC | FC | FC | LeNet-5 | LeNet-5 |
| | input size | 50 | 784 | 28 | $28 \times 28 \times 1$ | $32 \times 32 \times 3$ |
| | hidden layer size | 30 | 300 | 300 | $\mathrm{C}\,(5 \times 5) \times 20 - \mathrm{MP}\,2$[4] | |
| | | | 100 | 300 | $\mathrm{C}\,(5 \times 5) \times 50 - \mathrm{MP}\,2$[4] | |
| | | | | 300 | 500 | |
| | output layer size | 1 | 10 | 1 | 10 | |
| $\beta$ | nudging strength | | | 0.1 | | |
| $\mathcal{L}$ | loss | | | MSE | | |
| | initial weights & biases | uniform[2] | | $\sim \mathcal{N}(\mu = 0, \sigma = 0.05)$ | | |
| $\eta_{w,b}$ [ms$^{-1}$] | learning rate | 0.25 | $128 \times 0.125$ | $64 \times 0.125$ | $128 \times 0.125$ | $64 \times 0.125$ |
| | layerwise $\eta$ factors[3] | – | 1, .2, .1 | 1, .2, .1, .1 | 1, .2, .1 | 1, .2, .2, .2, .2, .1 |
| **Training parameters** | | | | | | |
| d$t$ [ms] | temporal resolution | 0.001 | 0.01 | 0.1 | 0.1 | 0.1 |
| $T_{\mathrm{pres}}$ | presentation time | 1 dt | 100 dt | 20 dt | 100 dt | 50 dt |
| | | = 0.001 ms | = 1 ms | = 2 ms | = 10 ms | = 5 ms |
| | batch size | 1 | 512 | 128 | 512 | 128 |
| | # training epochs | – | | 100 | | |
| | # train samples | – | 50000 | 40000 | 50000 | 342000 |
| | # validation samples | – | 10000 | 10000 | 10000 | 18000 |
| | # test samples | – | 10000 | 10000 | 10000 | 40000 |
| | # seeds | – | 10 | 9 | 9 | 9 |

[1] By "hard sigmoid" we mean the piecewise linear function $\varphi(x)$ that is obtained clipping a rectified linear unit (ReLU) to $[0, 1]$

$$\varphi(x) = x\theta(x) - (x - 1)\theta(x - 1) = \begin{cases} 0 & \text{if } x \leq 0 \\ 1 & \text{if } x \geq 1 \\ x & \text{else} \end{cases} \quad , \quad \varphi'(x) = \begin{cases} 1 & \text{if } x \in [0, 1] \\ 0 & \text{else} \end{cases}$$

where $\theta(x)$ denotes the Heaviside step function.

[2] PyTorch defaults

[3] These factors scale the learning rate $\eta$ for each layer independently.

[4] C and MP indicate convolutional and max pooling layers, respectively.

Also, we used a ReLU function for the hidden layer activation of the ANNs instead of the hard sigmoid activation that was used for the LE simulations. Furthermore, the BP results for the HIGGS dataset were produced using different activation functions for both hidden and output layers, namely $\tanh$ and sigmoidal, respectively.

To perform the simulations without prospective coding shown in Fig. 2b we set $\tau^{\mathrm{r}} = 0\,\mathrm{ms}$, $\tau^{\mathrm{m}} = 10\,\mathrm{ms}$ and used a temporal resolution of $\mathrm{d}t = 1\,\mathrm{ms}$ to obtain reasonable presentation times of multiple $\tau^{\mathrm{m}}$ that allow for relaxation. That is to say $T_{\mathrm{pres}} = 200\,\mathrm{d}t = 20\,\mathrm{ms} = 20\,\tau^{\mathrm{m}}$ for the purple curve in Fig. 2 compared to presentation times of $T_{\mathrm{pres}} = 1\,\mathrm{ms} = 0.05\,\tau^{\mathrm{m}}$ used for the LE simulations that employ the prospective coding allowing for much smaller presentation times.

## 15.2 Parameters used for the experiments shown in Fig. 3 and Fig. 5

**Table 2:** Neuron, network and training parameters used to produce the results using the microcircuit architecture.

| Parameter name | | Fig. 3 | Fig. 5 |
|---|---|---|---|
| **Neuron parameters** | | | |
| $C_{\mathrm{m}}$ | | 1 | 1 |
| $E_{\mathrm{l}}$ | | 0 | 0 |
| $g_{\mathrm{l}}$ | $[\mathrm{ms}^{-1}]^1$ | 0.03 | 0.03 |
| $g^{\mathrm{bas}}$ | $[\mathrm{ms}^{-1}]$ | 0.1 | 0.1 |
| $g^{\mathrm{api}}$ | $[\mathrm{ms}^{-1}]$ | 0.06 | 0.06 |
| $g^{\mathrm{den}}$ | $[\mathrm{ms}^{-1}]$ | 0.1 | 0.1 |
| $g^{\mathrm{nudge, I}}$ | $[\mathrm{ms}^{-1}]$ | 0.06 | 0.06 |
| $g^{\mathrm{nudge, tgt}}$ | $[\mathrm{ms}^{-1}]$ | 0.06 | 0.06 |
| $\tau_{\mathrm{eff}}$ | $[\mathrm{ms}]$ | $5.26^2$ | $5.26^2$ |
| $\varphi(x)$ | | $\log[1 + \exp(x)]$ | $\log[1 + \exp(x)]$ |
| **Network parameters** | | | |
| size input | | 9 | 1 |
| size hidden layer | | 30 | 1 |
| size output layer | | 3 | 1 |
| selfpred. $\eta_1^{\mathrm{PP}}$ | $[\mathrm{ms}^{-1}]$ | – | 0 |
| selfpred. $\eta_2^{\mathrm{PP}}$ | $[\mathrm{ms}^{-1}]$ | – | 0 |
| selfpred. $\eta_1^{\mathrm{IP}}$ | $[\mathrm{ms}^{-1}]$ | – | 40 |
| selfpred. $\eta_1^{\mathrm{PI}}$ | $[\mathrm{ms}^{-1}]$ | – | 50 |
| training $\eta_1^{\mathrm{PP}}$ | $[\mathrm{ms}^{-1}]$ | $5\mathrm{d}t/T_{\mathrm{pres}}{}^3$ | 50 |
| training $\eta_2^{\mathrm{PP}}$ | $[\mathrm{ms}^{-1}]$ | $1\mathrm{d}t/T_{\mathrm{pres}}{}^3$ | 10 |
| training $\eta_1^{\mathrm{IP}}$ | $[\mathrm{ms}^{-1}]$ | $2\mathrm{d}t/T_{\mathrm{pres}}{}^3$ | 20 |
| training $\eta_1^{\mathrm{PI}}$ | $[\mathrm{ms}^{-1}]$ | $0^4$ | $0^4$ |
| weight init (uniform) | | $[-1, 1]$ | $[-1, 1]$ |
| **Training parameters** | | | |
| start in selfpred. state | | yes | no |
| train biases | | no | no |
| delay on target signal | | $1\mathrm{d}t^5$ | $1\mathrm{d}t^5$ |
| selfpred. epochs | | – | 3 |
| training epochs | | 1000 | 500 |
| $\mathrm{d}t$ | $[\mathrm{ms}]$ | 0.1 | 0.01 |
| $T_{\mathrm{pres}}$ | | $3\,\mathrm{d}t - 5000\,\mathrm{d}t$ | $100\,\mathrm{d}t$ |

[1] To keep the other variables unitless, except for dynamical time scales, conductances and learning rates need to have the unit 1 / ms.

[2] The effective time constant is calculated from the neurons conductances: $\tau_{\mathrm{eff}} = \frac{C_{\mathrm{m}}}{g_{\mathrm{l}} + g^{\mathrm{bas}} + g^{\mathrm{api}}}$.

[3] Learning rates are scaled with varying $T_{\mathrm{pres}}$.

[4] If the network is starting in or has previously learned the self-predicting state the weights $\boldsymbol{W}^{\mathrm{PI}}$ do not need to be adapted.

[5] As the effect of a change in the input signal needs as many timesteps as there are hidden layers to reach the top layer of the network, the target signal needs to be delayed relative to the input by this amount of time steps.

## 15.3 Parameters used for the experiments shown in Fig. 4

Parameters not mentioned here explicitly (batch size, number of training, validation and test samples, learning rates, mean and variance for initial weights and biases) were the same as for the LE experiments shown in Fig. 2b (cf. Table 1).

**Table 3:** Additional parameters needed to reproduce the experiments shown in Fig. 4

| Symbol | Parameter name | Fig. 4a | Fig. 4c |
|---|---|---|---|
| **Training parameters** | | | |
| $\mathrm{d}t$ | temporal resolution | $0.002\,\mathrm{ms} - 2.0\,\mathrm{ms}$ | $0.01\,\mathrm{ms},\, 0.05\,\mathrm{ms}^{1}$ |
| $T_{\mathrm{pres}}$ | presentation time | $100\,\mathrm{d}t$ | $20\,\mathrm{d}t - 1000\,\mathrm{d}t$ |
| $\tau^{\mathrm{s}}$ | synaptic time constant | – | $0\,\mathrm{ms} - 2.0\mathrm{ms}$ |
| | # seeds | 4 | 3 |
| **Noise parameters** | | | |
| $\sigma_{\tau^{\mathrm{m/r}}}$ | time constant width$^{2}$ | $0\,\tau^{\mathrm{m/r}}, 0.01\,\tau^{\mathrm{m/r}}, 0.2\,\tau^{\mathrm{m/r}}$ | – |
| $\sigma_{\xi}$ | noise width | – | $0.2\,r_{\max}$ |
| | target LPF | – | yes$^{3}$ |

[1] The smaller value was used for the simulations of the green datapoints while the bigger value was used to obtain the blue and yellow curves.

[2] Time constants were clipped, i.e., $\tau^{\mathrm{m/r}} + \xi \in [1, 1000]$, to exclude the unphysical case of them to become negative.

[3] In case of Fig. 4c, an additional LPF with time constant $\tau^{\mathrm{trg}} = N \times \tau^{\mathrm{s}}$ where $N = \#$ layers was applied to the target signal. However, this is just an approximation for the $N$ LPFs that are being applied to the input signal during a forward pass. Yet it helps to reduce "wrong learning" during the short relaxation phases introduced by the synaptic filtering and can be neglected in the limit $\tau^{\mathrm{s}} \to 0$.

# 16 Broader impact

The physical interpretation of our model not only offers a biologically plausible implementation of continuous-time, continuously active neuro-synaptic dynamics, but also outlines a specific path towards mixed-signal (analog/digital) in-silico implementation. Even with existing technologies, such neuromorphic systems harbor the potential of surpassing their biological archetypes with respect to both energy efficiency and speed [2]. In conjunction with the inherent ability of our framework to support the processing of continuous data streams, the reduced power consumption of such devices makes them a prime candidate for the construction of autonomous, embodied learning machines.

While obviously beneficial for research and commercial deployment, one should be aware that improved training efficiency carries the risk of deploying ever more intransparent models [3]. Furthermore, along with its obvious benefits, improved, and in particular autonomous AI entails a plethora of far-reaching societal consequences that are the subject of ongoing academic and public debate [4]. Future progress hence needs to be considered carefully and responsibly, and, in particular, properly reflected in public policy.

On the path towards understanding and replicating biological intelligence, a corollary benefit for the scientific community may emerge. Modern machine learning requires enormous amounts of compute, thus largely limiting cutting-edge developments to institutions with the corresponding resources. The envisioned bio-inspired yet also bio-transcendent hardware systems have the potential to drastically increase the overall efficiency of custom-designed computational platforms. The resulting decrease in operating costs could thus significantly expedite the democratization of AI research.