# OpenReview forum: "Latent Equilibrium: A unified learning theory for arbitrarily fast computation with arbitrarily slow neurons"
_NeurIPS.cc/2021/Conference — NeurIPS 2021 Oral_

### Official Review · Reviewer_ztYa · 2021-07-14

**Rating:** 8
**Confidence:** 3

**Summary:**

The paper introduces Latent Equilibrium, a new framework for inference and learning in networks of slow components.

Authors use the leaky integrator dynamics that characterize biological neurons to jointly derive disentangled neuron and synapse dynamics. The neuronal dynamics evolve along a constant-energy manifold, called the Latent Equilibrium.

They show that LE allows for performance gains after the beginning of training, when the disruptive effects of relaxation begin to dominate, as well as robustness of LE on a MNIST classification task.

**Limitations And Societal Impact:**

yes

**Main Review:**

This is a great paper, well-written and well-motivated, introducing a new, biologically-inspired framework for inference and learning in networks of slow components. Authors claim that LE allows to represent the true errors in the networks before reaching a fixed point, and alleviates a second form of catastrophic forgetting.

The first point is well-proven in the experiments, but I find that the second one is not. I believe authors should include a transfer-learning type experiment, where we see the mitigation of the forgetting effect. Also, in the different experiments, authors should compare to other biologically-inspired architectures that alleviate this effect (such as Dendritic Gated Networks (Krishnagopal et al., Cosyne 2021)).


**Needs Ethics Review:**

Yes

**Time Spent Reviewing:**

4

---

> ### Author Response · Authors · 2021-08-10
> **Response**
>
> Thanks a lot for your review and positive evaluation!
>
> > Authors claim that LE allows to represent the true errors in the networks before reaching a fixed point, and alleviates a second form of catastrophic forgetting. The first point is well-proven in the experiments, but I find that the second one is not.
>
> Unfortunately, we have caused some confusion by introducing the concept of a “secondary form of catastrophic forgetting”. In particular, we do not mean that in the sense of unlearning of a previously learnt task because of learning a new, different task. Rather, we mean to describe an effect of unlearning which occurs in a network of slow components due to a misrepresentation of errors during network relaxation phases. So even if the network was presented with an input that would be correctly classified in the relaxed state, such that weight changes would have to be zero, plasticity does occur before relaxation and thus partially destroys the previously learned weight configuration. Thus, this will result in synaptic weight changes even in perfectly trained networks (see Fig. 1c). We will clarify this point in the revised manuscript.

---

> > ### Comment · Reviewer_ztYa · 2021-08-17
> > **Response to author's rebuttal**
> >
> > Thank you for your response and clarification!
> > I appreciate that you will clarify this point in the revised manuscript.

---

### Official Review · Reviewer_BSPV · 2021-07-17

**Rating:** 7
**Confidence:** 4

**Summary:**

The authors propose a framework from which a neuron model with (slow) neural dynamics close to physical substrates can be derived  along with a plasticity rule for learning that is similar to backprop. The neuron model allows fast propagation of signals through the network in spite of the slow dynamics of the components, which avoids having to control the timing of learning to turn it on only after the signal has propagated. The authors perform empirical evaluations of networks based on this model on various tasks.

**Limitations And Societal Impact:**

The authors don't discuss the limitations of the model in any detail.

**Main Review:**

**Update**: Based on the authors' response, I have revised my score to an accept.

## Originality:
The proposed model is original to my knowledge. Although the term and concept of "prospective coding" is loosely similar to the one in (Brea et al. 2016) -- might be a good idea to clarify this in the text.  The derivation of the neuron dynamics and learning rules from the energy function is novel. The related work is adequately discussed and the differences are clear.

Brea, Johanni, et al. "Prospective coding by spiking neurons." PLoS computational biology 12.6 (2016): e1005003.

## Quality:
The derivations of the neuron model and learning rule are technically sound. The evaluation of the model is adequate -- the fact that that model works well even with the standard convnet architecture for CIFAR10 is impressive. It would have been interesting to see results with other popular architectures like resnets.

The analysis of the robustness of the model to substrate imperfections is a bit simplistic -- since the actual imperfections in analog neuromorphic hardware can add a lot more than just noise and mismatches between internal variables. But given that actual testing on hardware is impractical, it is somewhat useful to see the model's robustness to the cases considered.

I'm not clear what the takeway message is from Section 5, especially given the significant gap in biological plausibility described below.

The authors mention biological plausibility as a key component of their model and yet don't discuss the role of spiking in information transmission. This particular model assumes that the information in membrane potential is directly transmitted to downstream neurons -- which is not biologically realistic. The only alternative way of transmission I can think of would be through the firing rate of the neuron and this would introduce a pretty large lag in the information transmission, avoiding which is one of the primary goals of this paper. So in my opinion, this major caveat to the biological plausibility of the model should be stated clearly and prominently.

## Clarity:
The paper is reasonably well written and the derivations and experiments are sufficiently clear.

Some notes:
* l.106: It's not clear what the contrast is to the previous work.
* l.97: The neuronwise sum over energies doesn't seem to include the loss term.

Minor:
* l.101: Eqn. 3 should be Eqn. 4?

## Significance:
The model itself is very clever and elegant, and well suited for implementation on certain types of neuromorphic substrates. But the arguments about biological plausibility are very shaky as I mentioned before, and I don't see how any insights can be gained there through the model. To be fair, many of the other cited similar papers also ignore the issue of actual information propagation through spikes. But since this paper purports to solve the problem of fast propagation of information, this issue is a lot more relevant here than in other work.

**Time Spent Reviewing:**

6

---

> ### Author Response · Authors · 2021-08-10
> **Response**
>
> Thank you for your review and your encouraging evaluation of our model! In the following we  will address your comments point by point. We hope that this clarifies the context of our work, in particular with respect to biological implementations.
>
> > Although the term and concept of "prospective coding" is loosely similar to the one in (Brea et al. 2016) -- might be a good idea to clarify this in the text.
>
> We are indeed aware of the work by Brea et al. 2016, and we think it addresses a different challenge. Technically, the models are related in that certain neuronal dynamical variables are looking ahead in time. However, Brea et al. solve a different problem, namely TD learning. Their prospective synaptic learning rule is not aimed at undoing the dissipative dynamics of neuronal membranes, but rather at implementing the future discounted states required for TD learning. Thus, the resulting neuronal and synaptic dynamics are very different from ours. Moreover, they do not discuss the problem of credit assignment via error backpropagation, nor its implementation in cortical circuitry. However, given your feedback, we think it is useful to reference this work in our final manuscript and explain its relationship and differences to ours.
>
> > The analysis of the robustness of the model to substrate imperfections is a bit simplistic -- since the actual imperfections in analog neuromorphic hardware can add a lot more than just noise and mismatches between internal variables. But given that actual testing on hardware is impractical, it is somewhat useful to see the model's robustness to the cases considered.
>
> We agree that the substrate imperfections addressed in the manuscript are far from a quantitatively faithful embodiment of what one would expect on an actual analog/mixed-signal neuromorphic device, although we would argue that ”a bit simplistic” doesn’t quite do our approach justice. An exhaustive investigation with respect to all possible imperfections is, if possible at all, easily worth an entire manuscript in itself (see for example Petrovici, Vogginger, Müller, Breitwieser, Lundqvist, et al., 2014 [a1]) and therefore, as we hope you would agree, beyond the scope of this manuscript. In this manuscript we focused on two of the most salient and disruptive aspects: temporal noise in signal transmission and fixed-pattern noise arising during the manufacturing process. Clearly, this investigation is merely a first step towards actual physical realization of the model. To extend our current discussion and owing to the additional space afforded to us during the review, we have included a new paragraph about further limitations of possible in-silico implementations in the final version of the paper.
>
> > This particular model assumes that the information in membrane potential is directly transmitted to downstream neurons -- which is not biologically realistic. The only alternative way of transmission I can think of would be through the firing rate of the neuron and this would introduce a pretty large lag in the information transmission, avoiding which is one of the primary goals of this paper. So in my opinion, this major caveat to the biological plausibility of the model should be stated clearly and prominently.
>
> You identify correctly that in the vanilla model (Eqn. 4 and 5), the downstream voltage has to be communicated to the upstream neuron to calculate its local error and synaptic update, but might have overlooked the solution suggested in Section 5 (Fast computation and learning in cortical microcircuits). There, we demonstrate that a biological implementations in cortical microcircuits, using purely rate-based communication between neurons and a rate-based synaptic learning rule (Eqn. 7), does the job as advertised (cf. Fig. 3).
>
> > The authors mention biological plausibility as a key component of their model and yet don't discuss the role of spiking in information transmission.
>
> We further agree that the abstraction of neuronal communication to rates is just that: an abstraction. However, we disagree that the use of such an abstraction rules out any important contribution of our model to the theory of neuronal computation. In keeping with a long tradition of successful rate-based models, we believe our model to also provide some important pieces of the cortical computation and learning puzzle. In particular, the “relaxation problem” we are solving within our framework is present in all biologically plausible implementations of error backpropagation in the brain [5-11]. While we do have some ideas about an application of LE to spiking neurons, these would go beyond the scope of this manuscript.
>
> To make these points clearer, we have added a new “limitations” section, in which we specifically address the role of spikes in the context of our framework. Furthermore, we have made the biological implications of our model more accessible.
>
> > l.106: It's not clear what the contrast is to the previous work.
>
> In contrast to previous work (see [6,8]), we derive neuronal dynamics as extrema of an energy function (cf. eqn. 3) defined on prospective voltages rather than by gradient descent on the energy; we have expanded the discussion of this point in the revised manuscript.
>
> > l.101: Eqn. 3 should be Eqn. 4?
>
> Yes, we should have said “by plugging Eqn. 3 into Eqn. 4”. We will amend this in the revised manuscript.
>
> > l.97: The neuronwise sum over energies doesn't seem to include the loss term.
>
> You are right that only the first part of the energy function (eqn. 2), excluding the loss, can be interpreted as a sum over mismatch energies. We will clarify this in the revised manuscript.
>
> We hope that our answer clarifies your questions and convinces you that, while admittedly not having provided the ultimate mechanistic explanation of credit assignment in the brain, our framework does constitute a relevant step towards understanding crucial aspects of cortical information processing, and also holds significance for artificial, but brain-inspired approaches to computation and learning.
>
> [a1] Petrovici MA, Vogginger B, Müller P, Breitwieser O, Lundqvist M, et al. (2014) Characterization and Compensation of Network-Level Anomalies in Mixed-Signal Neuromorphic Modeling Platforms. PLOS ONE 9(10): e108590. https://doi.org/10.1371/journal.pone.0108590

---

> > ### Comment · Reviewer_BSPV · 2021-08-27
> > **Satisfied with authors response**
> >
> > Overall, I still find the model very interesting and clever. As long as the specific limitations and the details not considered are made clear (and hopefully also prominent) in the manuscript, as the authors have promised, I would be willing to change my score to an accept.
> >
> > >>    The analysis of the robustness of the model to substrate imperfections is a bit simplistic -- since the actual imperfections in analog neuromorphic hardware can add a lot more than just noise and mismatches between internal variables. But given that actual testing on hardware is impractical, it is somewhat useful to see the model's robustness to the cases considered.
> >
> > > We agree that the substrate imperfections addressed in the manuscript are far from a quantitatively faithful embodiment of what one would expect on an actual analog/mixed-signal neuromorphic device, although we would argue that ”a bit simplistic” doesn’t quite do our approach justice. An exhaustive investigation with respect to all possible imperfections is, if possible at all, easily worth an entire manuscript in itself (see for example Petrovici, Vogginger, Müller, Breitwieser, Lundqvist, et al., 2014 [a1]) and therefore, as we hope you would agree, beyond the scope of this manuscript. In this manuscript we focused on two of the most salient and disruptive aspects: temporal noise in signal transmission and fixed-pattern noise arising during the manufacturing process. Clearly, this investigation is merely a first step towards actual physical realization of the model. To extend our current discussion and owing to the additional space afforded to us during the review, we have included a new paragraph about further limitations of possible in-silico implementations in the final version of the paper.
> >
> > Yes I was not suggesting that this manuscript should consider all imperfections. Giving a reader a sense of how complex this area is and that this paper focuses on two that you think are the most important would be sufficient IMO.
> >
> > >>    This particular model assumes that the information in membrane potential is directly transmitted to downstream neurons -- which is not biologically realistic. The only alternative way of transmission I can think of would be through the firing rate of the neuron and this would introduce a pretty large lag in the information transmission, avoiding which is one of the primary goals of this paper. So in my opinion, this major caveat to the biological plausibility of the model should be stated clearly and prominently.
> >
> > > You identify correctly that in the vanilla model (Eqn. 4 and 5), the downstream voltage has to be communicated to the upstream neuron to calculate its local error and synaptic update, but might have overlooked the solution suggested in Section 5 (Fast computation and learning in cortical microcircuits). There, we demonstrate that a biological implementations in cortical microcircuits, using purely rate-based communication between neurons and a rate-based synaptic learning rule (Eqn. 7), does the job as advertised (cf. Fig. 3).
> >
> > Right. My concern was that since the model specifically aims to tackle the response lag, it seems like switching to a rate based model re-introduces this problem? Or does the response lag introduced by the rate based model still not as bad as the other cases?
> >
> > >>    The authors mention biological plausibility as a key component of their model and yet don't discuss the role of spiking in information transmission.
> >
> > > We further agree that the abstraction of neuronal communication to rates is just that: an abstraction. However, we disagree that the use of such an abstraction rules out any important contribution of our model to the theory of neuronal computation. In keeping with a long tradition of successful rate-based models, we believe our model to also provide some important pieces of the cortical computation and learning puzzle. In particular, the “relaxation problem” we are solving within our framework is present in all biologically plausible implementations of error backpropagation in the brain [5-11]. While we do have some ideas about an application of LE to spiking neurons, these would go beyond the scope of this manuscript.
> >
> > I don't disagree at all that your model might potentially provide some important pieces of cortical computation, although I see that it might have come across that way from my comment. I just think the manuscript should also be very clear on the limitations vis a vis the details it does not consider and not position itself as actually providing a direct solution to the problem, more as providing a possible solution to one of the pieces in the puzzle.
> >
> > > To make these points clearer, we have added a new “limitations” section, in which we specifically address the role of spikes in the context of our framework. Furthermore, we have made the biological implications of our model more accessible.

---

> > > ### Author Response · Authors · 2021-09-22
> > > **Response to the reviewer**
> > >
> > > Thank you very much for the encouraging reply!
> > > We will definitely address the issues raised here in the revised version of the manuscript.

---

### Official Review · Reviewer_ob4X · 2021-07-21

**Rating:** 8
**Confidence:** 3

**Summary:**

The authors propose a scheme to reverse the deleterious effects of neuronal filtering / lags on inference and learning in neuronal networks. By formulating an energy function using the pre-filtered inputs to a neuron instead of the filtered inputs or outputs, they derive neuronal dynamics and plasticity rules involving pre-filtered inputs that overcome delays in inference and timing mismatches in plasticity leading to better learning.

**Limitations And Societal Impact:**

Yes

**Main Review:**

The authors propose a neat tweak to energy functions introduced in earlier works to derive neuronal dynamics and local plasticity rules that approximate backpropagation. Pre-filtered inputs are used in place of filtered inputs or outputs in the energy function bypassing the lag for purposes of inference and learning. This yields neuronal dynamics and plasticity rules that reflect evidence for 'prospective coding' in biological neurons and raise possibilities in biological plasticity. The authors are urged to connect / contrast this work to earlier work on prospective coding in spiking neurons by Brea et al 2016, and also expand on what predictions are made for biological synaptic plasticity experiments and whether these are indeed not already observed. This work deserves wide dissemination since this simple tweak could enable considerable benefits to the community searching for biological and neuromorphic analogues / alternative to backprop, as well as to the neuroscience community in further probing prospective coding.

**Time Spent Reviewing:**

2

---

> ### Author Response · Authors · 2021-08-10
> **Response**
>
> Thanks a lot for your positive review and your constructive feedback!
>
> > The authors are urged to connect / contrast this work to earlier work on prospective coding in spiking neurons by Brea et al 2016
>
> We are indeed aware of the work by Brea et al. 2016, and we think it addresses a different challenge. Technically, the models are related in that certain neuronal dynamical variables are looking ahead in time. However, Brea et al. solve a different problem, namely TD learning. Their prospective synaptic learning rule is not aimed at undoing the dissipative dynamics of neuronal membranes, but rather at implementing the future discounted states required for TD learning. Thus, the resulting neuronal and synaptic dynamics are very different from ours. Moreover, they do not discuss the problem of credit assignment via error backpropagation, nor its implementation in cortical circuitry. However, given your feedback, we think it is useful to reference this work in our final manuscript and explain its relationship and differences to ours.
>
> > [...] expand on what predictions are made for biological synaptic plasticity experiments and whether these are indeed not already observed
>
> We agree that our tweak to the energy functions carries important implications for synaptic plasticity. One implication of our framework is that plasticity is (in principle) equanimous about how fast the sensory input changes and learning is possible without the neuron itself ever reaching a steady state. We would argue that the ability of mammals to learn from a continuously, and often quickly changing input stream already provides at least a plausibility check, if not already some initial evidence for our hypothesis. We are already also considering more precise experimental investigations, for which we would predict that a controlled periodic input stream with near-arbitrarily short stimulus presentation times would show little impact on the ability of plasticity to instill high functional performance on the associated task. Thanks to the additional space afforded to us for the revised manuscript, we will address these ideas in the discussion section.

---

### Official Review · Reviewer_DcW4 · 2021-07-24

**Rating:** 9
**Confidence:** 4

**Summary:**

The authors introduce a novel framework called Latent Equilibrium that enables quasi-instantaneous inference regardless of network depth and learning without phased plasticity in networks with slow neuronal elements.  Notably, the framework offers a biologically plausible model of back-propagation. Authors show that the models generated according to the framework have reasonable prediction quality on standard benchmark datasets. The authors also show that the framework can generate good models of cortical networks. Finally, authors explore heterogeneous substrates and conclude that framework is robust.

**Limitations And Societal Impact:**

Yes

**Main Review:**

The LE framework and models presented in this manuscript are very novel, in particular the idea of “prospective” coding, which has no precedence in the literature as far as I am aware of. There are no obvious flaws that I can see in the theory and experiments presented in this paper. The paper is written clearly enough and provides enough experimental evidence to back the main claims introduced by the authors. The framework presented in this paper introduces a convincing proposal of a biologically-plausible implementation of the back-propagation algorithm, which is a fundamental concept in Deep Learning.  Therefore, I consider that the significance and potential impact of this paper is high.

**Time Spent Reviewing:**

2 hours

---

> ### Author Response · Authors · 2021-08-10
> **Response**
>
> Thank you very much for your positive feedback!

---

### Review · Ethics_Reviewer_QDKZ · 2021-08-03

**Recommendation:** See above.

**Ethics Review:**

This paper poses the same ethical issues as many other machine learning papers and their general quest to increase the efficiency of autonomous machines. The paper summarizes some of these concerns in the broader impacts statement. E.g., they note that "While obviously beneficial for research and commercial deployment, one should be aware that improved training efficiency carries the risk of deploying ever more intransparent models."

I'm not sure if one can say "obviously beneficial" but the assumption of benefits is one that is widespread in almost all of the papers submitted to NeurIPS and not specific to this paper.

---

### Review · Ethics_Reviewer_6ELm · 2021-08-11

**Recommendation:**

It is possible to address the minor concerns mentioned in the current version of the paper.

**Ethics Review:**

This paper does not in whole present significant ethical concerns.

One concern includes the use of the standard CIFAR-10 benchmark, based on the TinyImages dataset now removed from the web.  As the CIFAR-10 benchmark does not intentionally include images of people, it avoids some of the most significant issues raised by TinyImages dataset.  However, issues related to data owners consent for research use, for example, still remain.

The paper briefly mentions benefits related to power efficiency.  This may be tied to broader societal benefits related to the need for reducing energy usage and climate change impacts.

---

### Decision · Program_Chairs · 2021-09-28

**Decision:**

Accept (Oral)

**Comment:**

Reviewers were in agreement the proposed framework provided a novel and elegant model for learning in neurons with physical delays, with clear writing and presentation, and thorough experimental results.


**Consistency Experiment:**

NeurIPS has a long history of experimentation. In 2014, NeurIPS ran an experiment in which 10% of submissions were reviewed by two independent committees to quantify the randomness in the review process. This year, we repeated a variant of this experiment to see how the quality of the review process has changed over time.  This paper was part of the experiment and was therefore assigned to two committees (consisting of reviewers, an Area Chair, and a Senior Area Chair) that reached independent decisions.  If both committees made the same recommendation, this recommendation was followed. If a single committee recommended acceptance, the paper was accepted (with the exception of a few cases in which the other committee identified what we considered a fatal flaw, e.g., an error in a key result).

This copy’s committee reached the following decision: **Accept (Oral)**

The other committee assigned to the paper recommended **Accept (Poster)**.  You can find the other set of reviews, along with any follow up discussion with the authors here:
https://openreview.net/forum?id=IVV1putQ90